# MoH: Multi-Head Attention as Mixture-of-Head Attention

## Abstract

In this work, we upgrade the multi-head attention mechanism, the core of the Transformer model, to improve efficiency while maintaining or surpassing the previous accuracy level. We show that multi-head attention can be expressed in the summation form. Drawing on the insight that not all attention heads hold equal significance, we propose Mixture-of-Head attention (MoH), a new architecture that treats attention heads as experts in the Mixture-of-Experts (MoE) mechanism. MoH has two significant advantages: First, MoH enables each token to select the appropriate attention heads, enhancing inference efficiency without compromising accuracy or increasing the number of parameters. Second, MoH replaces the standard summation in multi-head attention with a weighted summation, introducing flexibility to the attention mechanism and unlocking extra performance potential. Extensive experiments on ViT, DiT, and LLMs demonstrate that MoH outperforms multi-head attention by using only 50%~90% of the attention heads. Moreover, we demonstrate that pre-trained multi-head attention models, such as LLaMA3-8B, can be further continue-tuned into our MoH models. Notably, MoH-LLaMA3-8B achieves an average accuracy of 64.0% across 14 benchmarks, outperforming LLaMA3-8B by 2.4% by utilizing only 75% of the attention heads. We believe the proposed MoH is a promising alternative to multi-head attention and provides a strong foundation for developing advanced and efficient attention-based models. The code and pre-train weights will be made available upon publication.

## 1 Introduction

Since attention is introduced and becomes a fundamental component of Transformers (Vaswani et al., 2017), multi-head attention has been the standard architecture for natural language processing (Kenton & Toutanova, 2019) and computer vision tasks (Dosovitskiy et al., 2021). It is well known that using multiple heads can improve model accuracy. However, not all attention heads hold equal significance. Some works have shown that many attention heads can be pruned without affecting accuracy. For example, Voita et al. (2019) introduces a method to quantify the usefulness of each attention head and prune those that are redundant. Similarly, Michel et al. (2019) challenges the necessity of multiple heads by examining the impact of extensive pruning across various settings. In computer vision, some works also identify attention head redundancy. Bhattacharyya et al. (2023) reduces redundancy to boost performance, while Yun & Ro (2024) develop single-head attention for efficiency. These findings demonstrate that vanilla multi-head attention contains redundant attention heads.

Besides, in multi-head attention, each attention head operates in parallel, and the final output is the sum of all attention heads (please refer to Section 3.1). Given that these attention heads operate independently and some may be redundant, we argue that it is possible to build a dynamic attention-head routing mechanism. Such a mechanism would enable each token to adaptively select the appropriate attention heads, enhancing inference efficiency without compromising accuracy.

To this end, we introduce Mixture-of-Head attention (MoH), a new architecture that integrates multi-head attention with the Mixture-of-Experts (MoE) mechanism (Jacobs et al., 1991). Specifically, we propose to treat attention heads as experts within the MoE framework. Similar to MoE, MoH consists of multiple attention heads and a router that activates the Top-K heads for each token. Moreover, we replace the standard summation in multi-head attention with a weighted summation. This design offers two significant advantages: **First**, MoH allows each token to select the most relevant attention

heads, improving inference efficiency without sacrificing accuracy or increasing the parameters. **Second**, by replacing the standard summation in multi-head attention with a weighted summation, MoH enhances the flexibility of the attention mechanism and increases the performance potential. Moreover, to efficiently capture common knowledge across different contexts, we designate a subset of attention heads as shared heads that remain always activated.

We evaluate our proposed MoH across various popular model frameworks, including Vision Transformers (ViT) (Dosovitskiy et al., 2021) for image classification, Diffusion models with Transformers (DiT) (Peebles & Xie, 2023) for class-conditional image generation, and Large Language Models (LLMs) (Brown et al., 2020; OpenAI, 2022; Ouyang et al., 2022) for language tasks. We show that MoH achieves competitive performance, or even outperforms multi-head attention with only 50%∼90% of the attention heads. For example, MoH-ViT-B achieves 84.9%/84.7% Top-1 accuracy on the ImageNet-1K (Deng et al., 2009) classification benchmark, surpassing well-tuned multi-head attention baselines with only 75%/50% of the attention heads.

Furthermore, we demonstrate that pre-trained multi-head attention models, such as LLaMA3-8B (Dubey et al., 2024), can be further continue-tuned into our MoH models. Specifically, using only about 3% (400B tokens) of the original LLaMA3 pre-training data for continue-tuning, MoH-LLaMA3-8B achieves an average accuracy of 64.0% across 14 benchmarks, outperforming LLaMA3-8B by 2.4% by utilizing only 75% of the attention heads. These results show that MoH is a promising alternative to vanilla multi-head attention, laying a solid foundation for developing advanced and efficient attention-based models. The main contributions are summarized as follows:

- We propose a dynamic attention-head routing mechanism that allows each token to adaptively select the appropriate attention heads, enhancing model performance and inference efficiency without increasing the number of parameters.

- In addition to training from scratch, we demonstrate that pre-trained multi-head attention models, such as LLaMA3-8B, can be further continue-tuned into our MoH models, greatly enhancing the applicability of the proposed MoH method.

- A wide range of experiments across various popular model frameworks, including ViT, DiT, and LLMs, confirm that MoH is a promising alternative to vanilla multi-head attention, laying a solid foundation for developing advanced and efficient attention-based models.

## 2 RELATED WORK

**Multi-Head Attention.** Transformers (Vaswani et al., 2017) have garnered significant interest and success in both natural language processing and computer vision. The success of transformers has been long attributed to the multi-head attention mechanism (Cordonnier et al., 2020). Multi-head attention mechanism is proposed by Vaswani et al. (2017) to enhance the representation power of an attention layer by allowing multiple attention heads to operate on different low-dimensional projections of the input. The outputs from these heads are then concatenated to form the final result.

**Mixture-of-Experts Models.** The Mixture-of-Experts (MoE) method (Du et al., 2022; Lewis et al., 2021; Rajbhandari et al., 2022; Roller et al., 2021; Zhou et al., 2022) is introduced to expand the capacity of deep neural networks without increasing computational costs. In this approach, only a subset of parameters, known as experts, is activated for each input. Shazeer et al. (2017) first introduces an MoE layer between LSTM layers. Switch Transformer (Fedus et al., 2022) further simplifies the gating mechanism by selecting only the Top-1 expert per token. Gshard (Lepikhin et al., 2021) improves the Top-2 expert routing strategy. In contrast to MoE, which emphasizes efficient parameter scaling while maintaining manageable computational costs, the proposed MoH focuses on reducing the activation of redundant attention heads without increasing the number of parameters.

## 3 METHODOLOGY

In this work, we aim to reduce the activation of redundant attention heads without increasing the number of parameters. A high-level comparison between the vanilla multi-head attention and our proposed Mixture-of-Head attention (MoH) is presented in Fig. 1.

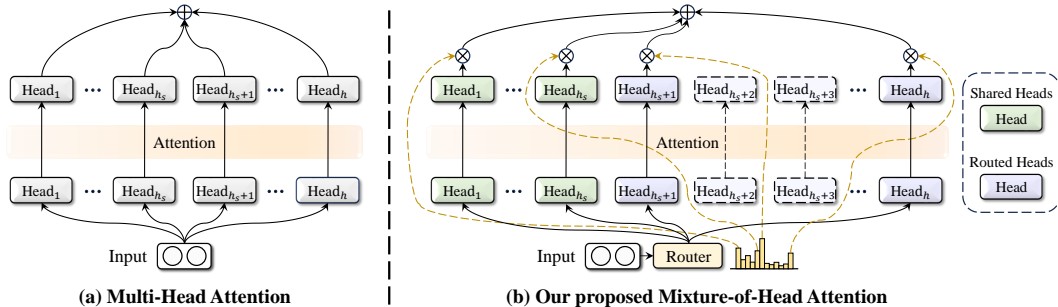

Figure 1: **A high-level comparison between the multi-head attention and our proposed mixture-of-head attention.** Subfigure (a) illustrates a standard multi-head attention layer with $h$ attention heads, while subfigure (b) demonstrates the Mixture-of-Head attention (MoH) architecture. It is important to note that MoH does not increase the number of attention heads, ensuring that the total parameter for MoH is comparable to that of the multi-head attention.

### 3.1 MULTI-HEAD ATTENTION

We begin by reviewing the standard multi-head attention mechanism introduced by Vaswani et al. (2017). The multi-head attention mechanism is based on scaled dot-product attention. Specifically, for $T$ tokens $\boldsymbol{X} \in \mathbb{R}^{T \times d_{in}}$ of $d_{in}$ dimensions each and $T'$ tokens $\boldsymbol{X}' \in \mathbb{R}^{T' \times d_{in}}$ of $d_{in}$ dimensions each, the scaled dot-product attention is computed as follows:

$$\text{Attention}(\boldsymbol{Q}, \boldsymbol{K}, \boldsymbol{V}) = \text{Softmax}\Big(\frac{\boldsymbol{Q}\boldsymbol{K}^\top}{\sqrt{d_k}}\Big)\boldsymbol{V},$$
$$\boldsymbol{Q} = \boldsymbol{X}\boldsymbol{W}_Q, \boldsymbol{K} = \boldsymbol{X}'\boldsymbol{W}_K, \boldsymbol{V} = \boldsymbol{X}'\boldsymbol{W}_V, \tag{1}$$

where $\boldsymbol{W}_Q \in \mathbb{R}^{d_{in} \times d_k}$, $\boldsymbol{W}_K \in \mathbb{R}^{d_{in} \times d_k}$, and $\boldsymbol{W}_V \in \mathbb{R}^{d_{in} \times d_v}$ represent the projection matrices for the query, key, and value, respectively. In self-attention, the input tokens are the same, i.e., $\boldsymbol{X}' = \boldsymbol{X}$, and it is common for the key and value dimensions to be equal, i.e., $d_v = d_k$.

**Concatenation Form.** To enhance the representation power of the attention layer, Vaswani et al. (2017) proposes to allow multiple attention heads to operate on different low-dimensional projections of the input tokens. Specifically, the multi-head attention mechanism computes $h$ different low-dimensional projections of $(\boldsymbol{Q}, \boldsymbol{K}, \boldsymbol{V})$, performs scaled dot-product attention for each head, concatenates the results, and applies a final projection to the concatenated output. The concatenation form of the multi-head attention can be formulated as:

$$\text{MultiHead}(\boldsymbol{X}, \boldsymbol{X}') = \text{Concat}(\boldsymbol{H}^1, \boldsymbol{H}^2, ..., \boldsymbol{H}^h)\boldsymbol{W}_O,$$
$$\boldsymbol{H}^i = \text{Attention}(\boldsymbol{X}\boldsymbol{W}_Q^i, \boldsymbol{X}'\boldsymbol{W}_K^i, \boldsymbol{X}'\boldsymbol{W}_V^i), \tag{2}$$

where $\boldsymbol{W}_Q^i \in \mathbb{R}^{d_{in} \times d_k/h}$, $\boldsymbol{W}_K^i \in \mathbb{R}^{d_{in} \times d_k/h}$, and $\boldsymbol{W}_V^i \in \mathbb{R}^{d_{in} \times d_v/h}$ represent the $i_{th}$ projection matrices for the query, key, and value, respectively. $\boldsymbol{W}_O \in \mathbb{R}^{d_v \times d_{out}}$ is the final projection matrix.

**Summation Form.** The multi-head attention mechanism is typically represented in its concatenation form. However, from another perspective, if we decompose $\boldsymbol{W}_O \in \mathbb{R}^{d_v \times d_{out}}$ by rows, we can express multi-head attention in a summation form. Specifically, $\boldsymbol{W}_O$ can be divided into $h$ matrices by rows, i.e., $[\boldsymbol{W}_O^1, \boldsymbol{W}_O^2, ..., \boldsymbol{W}_O^h] = \boldsymbol{W}_O$, where $\boldsymbol{W}_O^i \in \mathbb{R}^{d_v/h \times d_{out}}$. Finally, the summation form of the multi-head attention can then be formulated as:

$$\text{MultiHead}(\boldsymbol{X}, \boldsymbol{X}') = \sum_{i=1}^{h} \boldsymbol{H}^i \boldsymbol{W}_O^i. \tag{3}$$

The concatenation form can be viewed as a variant of the summation form, where the sum of the dimensions of all attention heads is exactly equal to the hidden size. As shown in Eq. 3, in standard multi-head attention, each attention head operates in parallel, and the final output is the sum of all attention heads. Since these attention heads function independently, we can build a dynamic attention-head routing mechanism allowing each token to adaptively select the most relevant attention heads, improving inference efficiency without compromising accuracy.

## 3.2 MIXTURE-OF-HEAD ATTENTION

Recently, the Mixture-of-Experts (MoE) method has emerged as a popular approach for scaling the parameters of large language models (Jiang et al., 2024). A typical MoE layer consists of multiple expert networks and a router that activates the Top-K experts. Generally, the number of activated experts $K$ is significantly smaller than the total number of experts to ensure inference efficiency.

**Heads as Experts.** Inspired by the great success of MoE, we propose Mixture-of-Head attention (MoH), which treats attention heads as experts. Specifically, MoH consists of $h$ heads $\boldsymbol{H} = \{H^1, H^2, ..., H^h\}$ and a router that activates the Top-K heads. Formally, given input tokens $\boldsymbol{X}$ and $\boldsymbol{X}'$, the output of MoH is the weighted sum of outputs from the $K$ selected heads:

$$\text{MoH}(\boldsymbol{X}, \boldsymbol{X}') = \sum_{i=1}^{h} g_i \boldsymbol{H}^i \boldsymbol{W}_O^i, \tag{4}$$

where $g_i$ represents the routing score. $g_i$ is non-zero only when the $i_{th}$ attention head is activated. This design provides two key advantages: On the one hand, MoH enables each token to select the most relevant attention heads, boosting inference efficiency while maintaining accuracy. On the other hand, in contrast to the standard summation in multi-head attention, the weighted summation in MoH enhances the flexibility of the attention mechanism and unlocks performance potential.

**Shared Heads.** In attention mechanism, some attention heads may capture common knowledge across different contexts, such as grammatical rules in language. Inspired by Dai et al. (2024), we designate a subset of heads as shared heads that remain always activated. By consolidating common knowledge within shared heads, we reduce redundancy among the other dynamically routed heads.

**Two-Stage Routing.** Moreover, to dynamically balance the weights between shared and routed heads, we propose a two-stage routing strategy. In this routing strategy, the routing scores are determined by both the score of each individual head and the score associated with the head type. Specifically, given the $t_{th}$ input token $\boldsymbol{x}_t \in \mathbb{R}^{d_{in}}$ in $\boldsymbol{X} \in \mathbb{R}^{T \times d_{in}}$, the routing score $g_i$ is defined as:

$$g_i = \begin{cases} \alpha_1 \text{Softmax}(\boldsymbol{W}_s \boldsymbol{x}_t)_i, & \text{if } 1 \le i \le h_s, \\ \alpha_2 \text{Softmax}(\boldsymbol{W}_r \boldsymbol{x}_t)_{i-h_s}, & \text{if } (\boldsymbol{W}_r \boldsymbol{x}_t)_{i-h_s} \in \text{Top-K}\big(\{(\boldsymbol{W}_r \boldsymbol{x}_t)_{i-h_s} | h_s + 1 \le i \le h\}\big), \\ 0, & \text{otherwise}, \end{cases} \tag{5}$$

where $h_s$ denotes the number of shared heads. $\boldsymbol{W}_s \in \mathbb{R}^{h_s \times d_{in}}$ and $\boldsymbol{W}_r \in \mathbb{R}^{(h-h_s) \times d_{in}}$ represent the projection matrices for the shared and routed heads, respectively. The coefficients $\alpha_1$ and $\alpha_2$ balance the contributions of the shared and routed heads, and are defined as:

$$[\alpha_1, \alpha_2] = \text{Softmax}(\boldsymbol{W}_h \boldsymbol{x}_t), \tag{6}$$

where $\boldsymbol{W}_h \in \mathbb{R}^{2 \times d_{in}}$ is the trainable projection matrix, and $d_{in}$ is the hidden size of $\boldsymbol{x}_t$.

**Load Balance Loss** Directly training an MoE layer often causes the majority of tokens to be routed to a small number of experts, leaving the remaining experts insufficiently trained (Shazeer et al., 2017). To avoid the unbalanced load in the proposed MoH, following previous MoE methods (Lepikhin et al., 2021; Wei et al., 2024), we apply a load balance loss. Specifically, for the $t_{th}$ input token $\boldsymbol{x}_t \in \mathbb{R}^{d_{in}}$ in $\boldsymbol{X} \in \mathbb{R}^{T \times d_{in}}$, the load balance loss $\mathcal{L}_b$ is formulated as:

$$\mathcal{L}_b = \sum_{i=h_s+1}^{h} f_i P_i, \ f_i = \frac{1}{T} \sum_{t=1}^{T} \mathbb{1}(\text{Token } \boldsymbol{x}_t \text{ selects Head } i), \ P_i = \frac{1}{T} \sum_{t=1}^{T} \text{Softmax}(\boldsymbol{W}_r \boldsymbol{x}_t)_{i-h_s}, \tag{7}$$

where $T$ denotes the number of tokens. $\mathbb{1}(*)$ denotes the indicator function.

**Total Training Objective.** It is worth noting that the MoH is a general framework. Therefore, we evaluate our proposed MoH across various popular model frameworks, including Vision Transformers (ViT), Diffusion models with Transformers (DiT), and Large Language Models (LLMs). Depending on the specific task, we require the task-specific loss. Finally, the total training loss is the weighted sum of the task-specific loss $\mathcal{L}_{task}$ and the load balance loss $\mathcal{L}_b$:

$$\mathcal{L} = \mathcal{L}_{task} + \beta \mathcal{L}_b, \tag{8}$$

where $\beta$ is the trade-off hyper-parameter to mitigate the risk of routing collapse. By default, the weight $\beta$ for the load balance loss is set to 0.01 for all tasks.

Table 1: **Comparisons to current state-of-the-art methods on ImageNet-1K classification.** All models are trained exclusively on the ImageNet-1K training set. Our MoH-ViT models, based on TransNeXt (Shi, 2024), are trained for 300 epochs using a resolution of 224×224. To ensure a fair comparison, we only replace the standard multi-head attention with our Mixture-of-Head attention (MoH), keeping all other training parameters identical to TransNeXt.

| Methods | #Params (M) | #Activated Heads (%) | Acc (%) | Methods | #Params (M) | #Activated Heads (%) | Acc (%) |
|---|---|---|---|---|---|---|---|
| DeiT-S (Touvron et al., 2021) | 22 | 100 | 79.8 | DeiT-B (Touvron et al., 2021) | 86 | 100 | 81.8 |
| T2T-ViT-19 (Yuan et al., 2021) | 39 | 100 | 81.9 | T2T-ViT-24 (Yuan et al., 2021) | 64 | 100 | 82.3 |
| Swin-S (Liu et al., 2021) | 50 | 100 | 83.1 | Swin-B (Liu et al., 2021) | 88 | 100 | 83.5 |
| PVTv2-B3 (Wang et al., 2022) | 45 | 100 | 83.2 | PVTv2-B5 (Wang et al., 2022) | 82 | 100 | 83.8 |
| CoAtNet-1 (Dai et al., 2021) | 42 | 100 | 83.3 | Focal-B (Yang et al., 2021) | 90 | 100 | 83.8 |
| Focal-S (Yang et al., 2021) | 51 | 100 | 83.5 | FocalNet-B (Yang et al., 2022b) | 89 | 100 | 83.9 |
| FocalNet-S (Yang et al., 2022b) | 50 | 100 | 83.5 | CoAtNet-2 (Dai et al., 2021) | 75 | 100 | 84.1 |
| MViTv2-S (Li et al., 2022) | 35 | 100 | 83.6 | MViTv2-B (Li et al., 2022) | 52 | 100 | 84.4 |
| UniFormer-B (Li et al., 2023b) | 50 | 100 | 83.9 | MOAT-2 (Yang et al., 2022a) | 73 | 100 | 84.7 |
| CAFormer-S36 (Yu et al., 2023) | 39 | 100 | 84.5 | iFormer-L (Si et al., 2022) | 87 | 100 | 84.8 |
| TransNeXt-S (Shi, 2024) | 50 | 100 | **84.7** | TransNeXt-B (Shi, 2024) | 90 | 100 | 84.8 |
| **MoH-ViT-S** | 50 | 80 | **84.7** | **MoH-ViT-B** | 90 | 75 | **84.9** |
| **MoH-ViT-S** | 50 | 75 | 84.6 | **MoH-ViT-B** | 90 | 50 | 84.7 |

## 4 EXPERIMENTS

### 4.1 VIT FOR IMAGE CLASSIFICATION

**Model Settings.** For Vision Transformers (ViT) (Dosovitskiy et al., 2021), our MoH-ViT models are implemented based on the TransNeXt (Shi, 2024) framework and trained from scratch on the ImageNet-1K dataset (Deng et al., 2009), which contains over 1.2 million images in 1,000 categories. To ensure a fair comparison, we only replace the standard multi-head attention with the proposed MoH, while keeping all other training parameters identical to TransNeXt.

**Training Details.** Our MoH-ViT models are trained for 300 epochs using automatic mixed precision across 8 GPUs. We follow the training strategy of TransNeXt, which includes various data augmentation techniques, including Random Augmentation (Cubuk et al., 2020), Mixup (Zhang, 2017), CutMix (Yun et al., 2019), and Random Erasing (Zhong et al., 2020). We also apply Label Smoothing (Szegedy et al., 2016) and DropPath (Huang et al., 2016) to regularize our models. We optimize our models using AdamW optimizer (Loshchilov & Hutter, 2017) with a gradient clipping norm of 1.0 and a weight decay of 0.05. The initial learning rate is set to 1e-3, with a 5-epoch warm-up starting at 1e-6. A cosine learning rate scheduler (Loshchilov & Hutter, 2016) is employed to decay the learning rate. During training, images are randomly cropped to a size of 224×224. It is worth noting that we do not use Exponential Moving Average (EMA) weights.

**Results.** As shown in Tab. 1, despite activating only a subset of attention heads, MoH-ViT achieves highly competitive performance compared to current state-of-the-art methods. For example, MoH-ViT-B achieves 84.9% Top-1 accuracy on the ImageNet-1K classification benchmark with just 75% of the attention head. In contrast, the well-established ViT baseline, TransNeXt, attains a slightly lower accuracy of 84.8% while requiring 100% of the heads to be activated. Tab. 1 demonstrates that MoH-ViT outperforms other models with fewer activated attention heads. This suggests that MoH is a promising alternative to vanilla multi-head attention for vision model design, offering the potential for competitive performance with more efficient attention head usage.

### 4.2 DIT FOR CLASS-CONDITIONAL IMAGE GENERATION

**Model Settings.** For Diffusion models with Transformers (DiT) (Peebles & Xie, 2023), we only replace the standard multi-head attention with our MoH in MoH-DiT models, while keeping all other training parameters identical to DiT. We use the ImageNet-1K dataset (Deng et al., 2009) for class-conditional image generation at a resolution of 256×256.

**Training Details.** Following DiT, the final linear layer is initialized with zeros, and all other layers follow standard ViT weight initialization. We train all models using the AdamW opti-

Table 2: **Comparisons to DiT on the benchmarking of class-conditional image generation on ImageNet-1K at 256×256 resolution.** To ensure a fair comparison, we only replace the standard multi-head attention with the MoH in MoH-DiT models, while keeping all other training parameters identical to DiT. "400K" denotes the training budget is 400K training steps.

| Methods | #Params (M) | #Activated Heads (%) | FID↓ | sFID↓ | IS↑ | Precision↑ | Recall↑ |
|---|---|---|---|---|---|---|---|
| DiT-S/2 400K (Peebles & Xie, 2023) | 33 | 100 | 68.40 | - | - | - | - |
| **MoH-DiT-S/2** 400K | 33 | 90 | **67.25** | **12.15** | **20.52** | **0.37** | **0.58** |
| **MoH-DiT-S/2** 400K | 33 | 75 | 69.42 | 12.85 | 19.96 | 0.36 | 0.55 |
| DiT-B/2 400K (Peebles & Xie, 2023) | 130 | 100 | 43.47 | - | - | - | - |
| **MoH-DiT-B/2** 400K | 131 | 90 | **43.40** | **8.40** | 33.51 | **0.49** | **0.63** |
| **MoH-DiT-B/2** 400K | 131 | 75 | 43.61 | 8.48 | 33.43 | **0.49** | 0.62 |
| DiT-L/2 400K (Peebles & Xie, 2023) | 458 | 100 | 23.33 | - | - | - | - |
| **MoH-DiT-L/2** 400K | 459 | 90 | **23.17** | **6.16** | **58.92** | **0.61** | **0.63** |
| **MoH-DiT-L/2** 400K | 459 | 75 | 24.29 | 6.38 | 57.75 | 0.60 | **0.63** |

Table 3: **Comparisons to current state-of-the-art methods on the benchmarking of class-conditional image generation on ImageNet-1K at 256×256 resolution.** "↑" denotes that higher is better. "↓" denotes that lower is better. "cfg" denotes the classifier-free diffusion guidance scale. We extend the training budget of our MoH-DiT-XL/2 to 7,000K training steps, aligning it with DiT-XL/2.

| Methods | #Activated Heads (%) | FID↓ | sFID↓ | IS↑ | Precision↑ | Recall↑ |
|---|---|---|---|---|---|---|
| ADM-G, ADM-U (Dhariwal & Nichol, 2021) | - | 3.94 | 6.14 | 215.84 | 0.83 | 0.53 |
| CDM (Ho et al., 2022) | - | 4.88 | - | 158.71 | - | - |
| LDM-8 (Rombach et al., 2022) | - | 15.51 | - | 79.03 | 0.65 | 0.63 |
| LDM-4 (Rombach et al., 2022) | - | 10.56 | - | 103.49 | 0.71 | 0.62 |
| LDM-4-G (cfg=1.25) | - | 3.95 | - | 178.22 | **0.81** | 0.55 |
| DiT-XL/2 7,000K (Peebles & Xie, 2023) | 100 | 9.62 | 6.85 | 121.50 | 0.67 | **0.67** |
| DiT-XL/2 7,000K (cfg=1.25) | 100 | 3.22 | 5.28 | 201.77 | 0.76 | 0.62 |
| **MoH-DiT-XL/2** 2,000K | 75 | 10.95 | 6.19 | 106.69 | 0.67 | 0.66 |
| **MoH-DiT-XL/2** 2,000K | 90 | 10.67 | 6.15 | 107.80 | 0.67 | 0.65 |
| **MoH-DiT-XL/2** 7,000K | 90 | 8.56 | 6.61 | 129.54 | 0.68 | **0.67** |
| **MoH-DiT-XL/2** 7,000K (cfg=1.25) | 90 | **2.94** | **5.17** | **207.25** | 0.77 | 0.63 |

mizer (Loshchilov & Hutter, 2017) with a constant learning rate of 1e-4, no weight decay, and a batch size of 256, applying horizontal flips for data augmentation. Following DiT, we employ the Exponential Moving Average (EMA) of MoH-DiT weights during training with a decay rate of 0.9999, generating all images using the EMA model. We use an off-the-shelf pre-trained variational autoencoder (Kingma, 2013) model from Stable Diffusion (Rombach et al., 2022). Following TransNeXt, our attention-head activation budget is unevenly distributed across layers, with fewer attention heads activated in the shallow layers and more in the deeper layers.

**Evaluation Benchmarks.** To evaluate generation performance, we use Frechet Inception Distance (**FID**) (Heusel et al., 2017) to assess overall sample quality, **Precision** and **Recall** (Kynkäänniemi et al., 2019) to measure fidelity and diversity separately, and **sFID** (Nash et al., 2021) as a metric that better captures spatial relationships than FID. Moreover, we use Inception Score (**IS**) (Salimans et al., 2016) as another metric for fidelity.

**Results.** To conduct comparative evaluations of our proposed MoH-DiT models against vanilla DiT models, we start with Small models and expand to XLarge models. As shown in Tab. 2, MoH-DiT models consistently outperform vanilla DiT models with 90% of attention heads activated. However, when only 75% of the attention heads are activated, MoH-DiT models perform worse than DiT models with 100% of attention heads activated. This may be because image generation tasks are dense prediction tasks that require attention mechanisms to capture pixel-level fine-grained relationships, leaving less redundancy in the attention heads compared to image classification tasks. Moreover, we extend the training budget of our MoH-DiT-XL/2 to 7,000K training steps, aligning it

Table 4: **Comparisons between MoH-LLMs and vanilla LLMs.** "100B" denotes a training budget of 100 billion tokens, while "200B" denotes a budget of 200 billion tokens. We observe that larger models, e.g., MoH-LLM-B, generally perform worse than smaller models, e.g., MoH-LLM-S, on TruthfulQA, consistent with the findings reported by Lin et al. (2022).

| Methods | #Activated Heads (%) | Language Tasks | | | | | | Avg. |
|---|---|---|---|---|---|---|---|---|
| | | SciQ | PIQA | WinoGrande | OpenbookQA | LogiQA | TruthfulQA | |
| LLM-S 100B | 100 | 63.0 | **63.1** | 51.1 | 27.4 | **26.9** | 31.6 | 43.9 |
| **MoH-LLM-S** 100B | 75 | 64.7 | 62.0 | 50.6 | 28.8 | 26.4 | 35.2 | 44.6 |
| **MoH-LLM-S** 100B | 50 | **67.0** | 62.2 | **51.5** | **29.2** | 26.7 | **35.6** | **45.4** |
| LLM-B 100B | 100 | 73.1 | **69.7** | 52.0 | **31.8** | **28.4** | 29.5 | 47.4 |
| **MoH-LLM-B** 100B | 75 | 74.7 | 69.2 | **52.8** | 30.0 | 28.1 | 32.2 | **47.8** |
| **MoH-LLM-B** 100B | 50 | **75.2** | 67.0 | 52.0 | 29.0 | 26.9 | **32.8** | 47.2 |
| LLM-B 200B | 100 | 73.1 | **70.3** | 53.3 | **32.4** | 29.0 | 29.5 | 47.9 |
| **MoH-LLM-B** 200B | 75 | **76.0** | 69.2 | 52.7 | 30.4 | **29.8** | 32.6 | **48.5** |
| **MoH-LLM-B** 200B | 50 | 75.6 | 66.9 | **53.5** | 29.4 | 26.7 | **32.7** | 47.5 |

with DiT-XL/2. As shown in Tab. 3, despite activating 90% attention heads, MoH-DiT-XL/2 achieves highly competitive performance compared to current state-of-the-art methods. These results suggest that MoH is a promising alternative to multi-head attention for diffusion models.

## 4.3 TRAINING LLMS FROM SCRATCH

**Model Settings.** For training LLMs from scratch, we use Megatron (Shoeybi et al., 2019), an open-source training code, as the training framework. Please refer to the Appendix for detailed hyper-parameter settings (Tab. A) of various MoH-LLMs. All models are trained with the AdamW optimizer (Loshchilov & Hutter, 2017), using a batch size of 4 million tokens with a sequence length of 2048. The final learning rate is set to 10% of the maximum. During training, a weight decay of 0.1 and gradient clipping of 1.0 are applied. For LLM-S and MoH-LLM-S, the maximum learning rate is set to 3e-4. For LLM-B and MoH-LLM-B, the maximum learning rate is set to 5e-4.

**Training Details.** We only use public datasets for training, ensuring accessibility for academic research. Specifically, we sample from the **RedPajama** (Computer, 2023), **Dolma** (Soldaini et al., 2024), and **Pile** (Gao et al., 2020) datasets according to different sampling probabilities. Please refer to the Appendix for detailed sample ratios (Tab. B). Following previous works, we utilize the tokenizer from LLaMA2 (Touvron et al., 2023), which contains 65,536 vocabulary tokens.

**Evaluation Benchmarks.** The evaluation is performed on multiple benchmarks using the Eleuther AI Language Model Evaluation Harness (Gao et al., 2024), a unified framework for testing generative language models. Since the parameters are only about 0.2B for the smallest model, we select 6 simple benchmarks as the metric. Specifically, we report 0-shot accuracy on **SciQ** (Welbl et al., 2017), **PIQA** (Bisk et al., 2020), **WinoGrande** (Sakaguchi et al., 2021), **OpenbookQA** (Mihaylov et al., 2018), **LogiQA** (Liu et al., 2020), and **TruthfulQA** (Lin et al., 2022).

**Results.** As shown in Tab. 4, despite activating only a subset of attention heads, MoH-LLMs achieve highly competitive performance compared to our baseline models. For example, MoH-LLM-S achieves an average accuracy of 45.4% with just 50% of the attention heads activated. In contrast, the baseline model reaches a slightly lower accuracy of 43.9% with 100% of the attention heads activated. These results suggest that MoH is a promising alternative to vanilla multi-head attention for training LLMs from scratch. Surprisingly, we find that for MoH-LLM-S, activating only 50% of the attention heads outperforms activating 75%. We consider it may be because when both the model and dataset are small, activating fewer heads effectively regularizes the model. However, as the amount of data increases, activating more heads offers a higher potential for performance.

## 4.4 CONTINUE-TUNING LLaMA3-8B

**Model Settings.** To significantly enhance the applicability of the proposed MoH method, we also attempt to further continue-tune pre-trained multi-head attention models, such as LLaMA3-8B, into MoH models. However, this presents three challenges. **(i) Determining the shared attention**

Table 5: **Comparisons between MoH-LLaMA3-8B and LLaMA3-8B.** Please refer to Tab. E in the Appendix for the performance of the model at the end of the first stage of training.

| Methods | #Activated Heads (%) | MMLU (5) | CEVAL (5) | CMMLU (5) | GSM8K(8) | TruthfulQA |
|---|---|---|---|---|---|---|
| LLaMA3-8B (Dubey et al., 2024) | 100 | 65.2 | 52.3 | 50.7 | 49.5 | 35.4 |
| **MoH-LLaMA3-8B** | 75 | 65.8 | 61.5 | 64.4 | 56.9 | 44.0 |

| Methods | #Activated Heads (%) | HellaSwag (10) | LogiQA | BoolQ (32) | LAMBADA | SciQ |
|---|---|---|---|---|---|---|
| LLaMA3-8B (Dubey et al., 2024) | 100 | 81.9 | 30.0 | 83.9 | 75.5 | 94.0 |
| **MoH-LLaMA3-8B** | 75 | 80.1 | 30.3 | 84.0 | 76.4 | 92.2 |

| Methods | #Activated Heads (%) | PIQA | WinoGrande | NQ (32) | ARC-C (25) | Average |
|---|---|---|---|---|---|---|
| LLaMA3-8B (Dubey et al., 2024) | 100 | 81.0 | 72.5 | 31.5 | 59.0 | 61.6 |
| **MoH-LLaMA3-8B** | 75 | 78.8 | 72.9 | 28.3 | 60.1 | **64.0** |

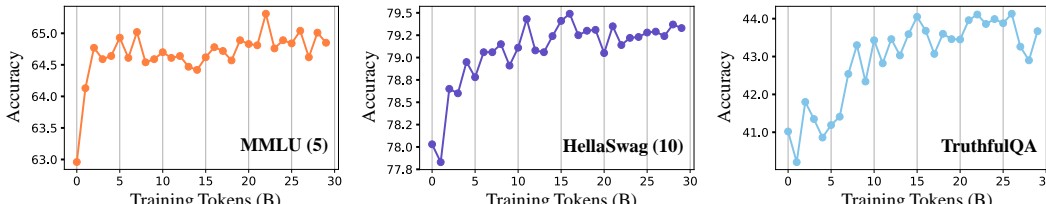

Figure 2: **Performance evolution during continue-tuning.** The MoH model quickly recovers to over 95% of the performance of the original model within a training budget of 10B tokens. Then, the performance gradually improves with the increase of the training tokens.

**heads**: We simply select the first 16 attention heads of each layer as shared heads. **(ii) Adding head routers**: Integrating a randomly initialized router into the pre-trained model without compromising its original performance requires careful training techniques. To address this, we propose a parameter-free router that determines routing scores using the $\ell_2$ norm of the query of each attention head. **(iii) Weighting attention heads**: We observe that weighting the attention head outputs significantly alters the distribution of the output of the attention layer, which necessitates a large amount of training data to restore the original performance. To tackle this, we quantize the routing score and use the straight-through estimator (Bengio et al., 2013; Liu et al., 2022) to back-propagate the gradients through the sparsity function. Specifically, given the input token $x$, we employ a quantizer for activation routing scores, with its forward pass formulated as:

$$g_i^q = \mathbb{1}(\text{Token } x \text{ selects Head } i), \tag{9}$$

where $\mathbb{1}(*)$ denotes the indicator function. $g_i^q$ represents the quantized routing score. We then adopt a straight-through estimator, which assigns the incoming gradients to a threshold operation to be the outgoing gradients, which is formulated as:

$$\frac{\partial \mathcal{L}}{\partial g_i^q} = \frac{\partial \mathcal{L}}{\partial g_i}, \tag{10}$$

where $g_i$ denotes the real-valued routing score. This simple approximation function significantly mitigates the issue of gradient vanishing (Wang et al., 2024). Similar to training LLMs from scratch, we also use Megatron (Shoeybi et al., 2019), an open-source training code, as the training framework.

**Training Details.** We find that if there is a discrepancy between the continue-training data and the original training data distribution of the model, the performance of the model may fluctuate wildly at the beginning of the training process. Since we are unable to have access to the raw training data of LLaMA3, we address these potential performance fluctuations by dividing the training process into two stages. In the first stage, we continue-tune the original LLaMA3-8B model using 300B tokens to adapt the model to our dataset. In the second stage, we continue-tune this adapted model into our proposed MoH model with 100B tokens. During the first stage, the maximum learning rate is set to

Table 6: **Ablation study on the impact of each component of the proposed MoH.** The image classification results are from MoH-ViT-S, by utilizing 75% of the attention heads with a training budget of 100 epochs. The class-conditional image generation results come from MoH-DiT-S/2-400K, also by using 75% of the attention heads, with a training budget of 400K training steps.

| Shared | Two-Stage | Image Classification | Class-Conditional Image Generation | | | | |
|--------|-----------|----------------------|------|-------|------|-----------|--------|
| Heads | Routing | Acc (%)↑ | FID↓ | sFID↓ | IS↑ | Precision↑ | Recall↑ |
| | | 75.6 | 71.97 | 13.58 | 19.06 | 0.35 | **0.55** |
| ✓ | | 78.3 | 69.54 | **12.80** | 19.67 | **0.36** | **0.55** |
| ✓ | ✓ | **78.6** | **69.42** | 12.85 | **19.96** | **0.36** | **0.55** |

Table 7: **Ablation study on the impact of the shared heads ratio among activated heads.** All results are from MoH-ViT-S, by using 75% of the heads with a training budget of 100 epochs.

| Ratio of Shared Heads | 13.9% | 27.6% | 31.3% | 35.9% | 37.5% | 40.5% | 46.8% | 60.4% | 74.0% |
|-----------------------|-------|-------|-------|-------|-------|-------|-------|-------|-------|
| Accuracy (%) | 78.6 | 78.5 | 78.4 | 78.4 | 78.5 | 78.6 | 78.4 | 78.6 | 78.4 |

6e-5, and the final learning rate is 6e-6. In the second stage, the maximum learning rate is set to 2e-5, and the final learning rate is 1e-6. For both stages, we employ the AdamW optimizer (Loshchilov & Hutter, 2017), with a batch size of 16 million tokens with a sequence length of 8192. During training, we use a weight decay of 0.1 and gradient clipping of 1.0.

**Evaluation Benchmarks.** We use the Eleuther AI Language Model Evaluation Harness (Gao et al., 2024) to evaluate models on multiple key benchmarks. Specifically, we utilize the lm-evaluation-harness package to assess performance on a comprehensive suite of downstream tasks: (i) Following Pythia (Biderman et al., 2023), we report 0-shot accuracy on **LAMBADA** (Paperno et al., 2016), **LogiQA** (Liu et al., 2020), **PIQA** (Bisk et al., 2020), **SciQ** (Welbl et al., 2017), and **WinoGrande** (Sakaguchi et al., 2021). (ii) We report the accuracy of Chinese tasks, including 5-shot **CEVAL** (Huang et al., 2023) and 5-shot **CMMLU** (Li et al., 2023a). (iii) We report the accuracy of tasks from the Open LLM Leaderboard (Beeching et al., 2023), including 10-shot **HellaSwag** (Zellers et al., 2019), 25-shot **ARC Challenge (ARC-C)** (Clark et al., 2018), and 5-shot **MMLU** (Hendrycks et al., 2021). (iv) We report the exact match score for 32-shot **Natural Questions (NQ)** (Kwiatkowski et al., 2019) and the accuracy for 32-shot **BoolQ** (Clark et al., 2019). (v) We report the exact match score for 8-shot **GSM8K** (Cobbe et al., 2021) to evaluate the math ability. (vi) Moreover, we report 0-shot accuracy on **TruthfulQA** (Lin et al., 2022) to assess the ability to generate truthful answers.

**Results.** As shown in Fig. 2, MoH-LLaMA3-8B quickly recovers to over 95% of the performance of the original model within a training budget of 10B tokens. After continue-tuning with 100B tokens, as shown in Tab. 5, MoH-LLaMA3-8B achieves an average accuracy of 64.0% across 14 benchmarks, outperforming LLaMA3-8B by 2.4% by utilizing only 75% of the attention heads. These results demonstrate that pre-trained multi-head attention models can be further continue-tuned into our MoH models, significantly enhancing the applicability of the MoH method.

## 4.5 ABLATIVE ANALYSIS

**Effect of Each Component of the Proposed MoH.** To explore the impact of each component of our MoH method, we provide the ablation results in Tab. 6. "Shared Heads" refers to a subset of attention heads that are always activated. "Two-Stage Routing" represents the dynamic coefficient that balances the weights between shared and routed heads over the routing score, as described in Eq. 5 and Eq. 6. As shown in Tab. 6, shared heads significantly improve model performance by effectively capturing common knowledge, allowing the routed heads to focus more on domain-specific information. Moreover, two-stage routing further enhances model performance by dynamically balancing the weights between shared and routed heads. Our full model achieves the best performance, demonstrating that both components significantly benefit the attention mechanism.

**Effect of the Shared Heads Ratio among Activated Heads.** In Tab. 7, we provide the ablation study on the shared heads ratio among activated heads. We find that model performance remains relatively consistent across a wide range of shared heads ratios (from 13.9% to 74.0%). These results indicate that the performance of the model is stable as long as the shared heads ratio is not extreme.

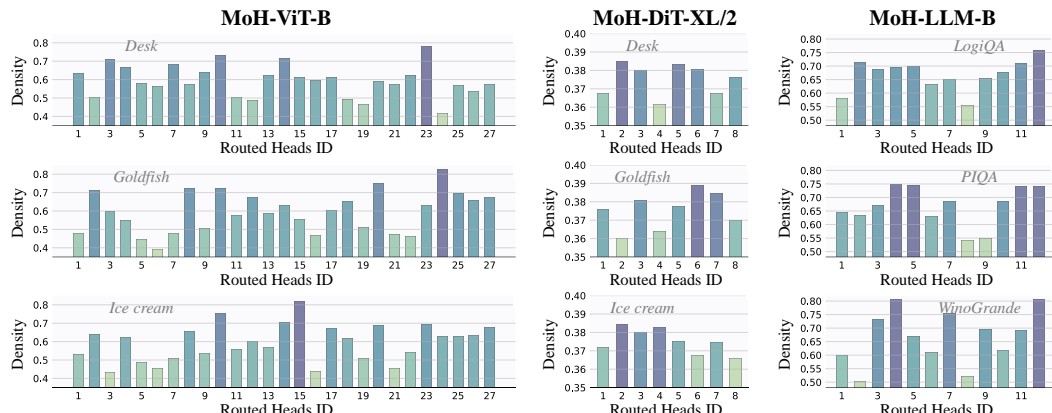

Figure 3: **Visualization of the head load distribution in the final MoH layer.** For ViT and DiT, we present the head load distributions for the categories "Desk", "Goldfish", and "Ice cream". For LLM, we display the head distributions for the tasks "LogiQA", "PIQA", and "WinoGrande". MoH-ViT-B, MoH-DiT-XL/2, and MoH-LLM-B activate 75%, 90%, and 75% of the attention heads, respectively. "Density" denotes the ratio of the number of head activations to the total number of tokens.

From another perspective, shared heads can be viewed as a form of Soft MoE (Puigcerver et al., 2024). Based on the findings from the Soft MoE paper (Puigcerver et al., 2024), we recommend using a higher ratio of shared heads among the activated heads (greater than 40%).

## 5 DISCUSSION

**Visualization of the Head Load Distribution.** As shown in Fig. 3, we observe significant variation in attention head assignments across different categories and task topics, indicating that the MoH model adapts to diverse tasks by employing distinct head assignment patterns. This characteristic of MoH allows different attention heads to focus on different types of tasks, making parameter utilization more efficient than multi-head attention. For additional visualizations of MoH-LLaMA3-8B and a detailed analysis of the head load distribution, please refer to Appendix D.

**The Difference between MoH and MoA.** We clarify the differences between MoH and MoA (Zhang et al., 2022) from the following three aspects. **First, in terms of motivation**, the goal of MoH is to improve the efficiency and performance of the attention mechanism without increasing the number of parameters. In contrast, MoA shares the motivation of MoE, which is to expand model parameters while keeping inference costs low. Therefore, the model settings of MoH are more stringent than those of MoA. **Second, in terms of methodology**, our MoH introduces shared heads and two-stage routing to enhance the standard MoE method. More importantly, we show that pre-trained multi-head attention models can be further continue-tuned into our MoH models, greatly improving the applicability of the proposed MoH method. In contrast, MoA directly combines multi-head attention with MoE. Due to the adoption of shared keys and values, MoA must be trained from scratch, which limits its applicability. **Finally, in terms of model frameworks**, our MoH is validated across various popular model frameworks and tasks, including ViT, DiT, and decoder-only LLMs, while MoA is only validated for language tasks.

## 6 CONCLUSION

In this paper, we introduce MoH, a promising alternative to multi-head attention. MoH enables each token to adaptively select the appropriate attention heads, improving both model performance and inference efficiency without increasing the number of parameters. Extensive experiments across various popular model frameworks, including ViT, DiT, and LLMs, demonstrate that MoH outperforms multi-head attention, even when using only 50%∼90% of the attention heads. More encouragingly, we show that pre-trained multi-head attention models, such as LLaMA3-8B, can be further continue-tuned into our MoH models, significantly enhancing the applicability of the proposed MoH method. This work represents a promising step toward advanced and efficient attention-based models, which may be meaningful and helpful to both the research and industrial communities.

## REPRODUCIBILITY STATEMENT

1. For model settings.
   (a) We outline the model settings in Section 4.
   (b) We describe in detail the MoH-LLM and MoH-LLaMA3-8B settings in Appendix B.
2. For training hyperparameters.
   (a) We outline the training hyperparameters in Section 4.
   (b) We describe in detail the training hyperparameters of MoH-LLM and MoH-LLaMA3-8B in Appendix B.
3. For code.
   (a) We have attached the code to the supplementary material.
   (b) We promise to release a more detailed and clean code version upon publication.
   (c) We will also release pre-train weights upon publication.

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

# A   Appendix

**Abstract.**   This appendix provides additional discussions (Appendix A), implementation details (Appendix B), several additional experiments (Appendix C), more qualitative analysis (Appendix D), and details of quantitative evaluations for LLMs (Appendix E).

**Code.**   We have attached the code to the supplementary material. In this code, we also provide the evaluation process of the proposed method. We promise to release a more detailed and clean code version upon publication.

## A   Additional Discussions

### A.1   Limitations and Future Work

In this section, we delineate the limitations of our work and outline avenues for future research.

**Heterogeneous Attention Heads.**   We find that different attention heads operate in parallel within the attention mechanism, suggesting that different heads can have varying hidden sizes. Future work could explore the use of heterogeneous attention heads based on our MoH framework.

**Lower Activation Rate.**   Currently, MoH outperforms multi-head attention by utilizing only 50%∼90% of the attention heads. However, this is still a relatively high proportion. Future work could aim to further optimize MoH, reducing head activation to less than 50%.

**Multimodal Inputs.**   Effectively processing information from multiple modalities in the attention mechanism remains an open question. Recent work (Wan et al., 2024) has shown that visual and textual tokens exhibit distinct attention patterns in multi-head attention. Future work could explore the attention patterns of MoH with different modal inputs, for example within multimodal large language models (Jin et al., 2024; Lin et al., 2023; 2024; Liu et al., 2024).

**More Downstream Tasks.**   We evaluate our proposed MoH across various popular model frameworks, including ViT for image classification, DiT for class-conditional image generation, and LLMs for language tasks. Future work can explore the application of MoH in more downstream tasks, such as audio tasks and multimodal tasks.

**More Parameters.**   Due to computational constraints, the maximum number of MoH model parameters in our experiments is limited to 8B (MoH-LLaMA3-8B). However, our MoH method is highly generalizable and can be scaled to larger models in future research.

## B   Implementation Details

### B.1   Training LLMs from Scratch

**Model Settings.**   For training LLMs from scratch, we use Megatron (Shoeybi et al., 2019), an open-source training code, as the training framework. The detailed hyper-parameter settings of various MoH-LLMs are shown in Tab. A.

Table A: **Sizes and architectures of MoH-LLMs and LLMs.** "MoH-LLM-B" has more parameters than "LLM-B" due to the additional parameters introduced by the router network.

| Methods | #Params | #Layers | #Hidden Size | #Intermediate Size | #Heads | #Head Dim |
|---------|---------|---------|--------------|--------------------|--------|-----------|
| LLM-S | 186 | 12 | 768 | 2048 | 12 | 64 |
| MoH-LLM-S | 186 | | | | | |
| LLM-B | 881 | 24 | 1536 | 4096 | 16 | 96 |
| MoH-LLM-B | 882 | | | | | |

**Data Details.**   Consistent with previous works, we use the tokenizer of LLaMA2, which contains 65,536 vocabulary tokens. It is worth noting that MoH-LLM is trained exclusively on public datasets, making it accessible for academic research settings. Tab. B shows the detailed sample ratios of

different open-source datasets. Specifically, we sample from the following datasets according to different sampling probabilities:

- The **RedPajama** (Computer, 2023) includes training data from seven domains: Common-Crawl, C4, Github, Wikipedia, Books, ArXiv, and StackExchange.

- The **Dolma** (Soldaini et al., 2024), a large and diverse open English text corpus, contains 3 trillion tokens sampled from seven sources, including web pages from Common Crawl, code from The Stack, curated web data from C4 (Raffel et al., 2020), social media conversations from Reddit, academic papers from PeS2o, public domain books from Project Gutenberg, and comprehensive content from Wikipedia and Wikibooks.

- The **Pile** (Gao et al., 2020), an open-source English text corpus for training large language models, includes 22 diverse, publicly available datasets such as Wikipedia, NIH ExPorter, ArXiv, Books3, BookCorpus2, OpenSubtitles, YoutubeSubtitles, and Enron Emails.

Table B: **Sampling ratio of different open-source datasets for MoH-LLMs.** MoH-LLM is trained exclusively on public datasets, making it accessible for academic research settings.

|  | Sampling Ratio |
| --- | --- |
| Redpajama Books | 4.24% |
| Redpajama Wikipedia | 3.50% |
| Redpajama ArXiv | 4.37% |
| Redpajama StackExchange | 3.19% |
| Redpajama C4 | 10.94% |
| Dolma | 61.28% |
| Pile | 12.48% |

**Training Hyper-Parameters.** Tab. C shows the detailed training hyper-parameters of MoH-LLMs. Specifically, all MoH-LLMs are trained with the AdamW optimizer (Loshchilov & Hutter, 2017), using a batch size of 4 million tokens with a sequence length of 2048. The final learning rate is set to 10% of the maximum. During training, a weight decay of 0.1 and gradient clipping of 1.0 are applied. For LLM-S and MoH-LLM-S, the maximum learning rate is set to 3e-4. For LLM-B and MoH-LLM-B, the maximum learning rate is set to 5e-4.

Table C: **Training hyper-parameters of MoH-LLMs.**

|  | **MoH-LLM-S** 100B (**LLM-S** 100B) | **MoH-LLM-B** 100B (**LLM-B** 100B) | **MoH-LLM-B** 200B (**LLM-B** 200B) |
| --- | --- | --- | --- |
| Training budget | 100B | 100B | 200B |
| Maximum learning rate | 3e-4 | 5e-4 | 5e-4 |
| Final learning rate | 3e-5 | 5e-5 | 5e-5 |
| LR warmup init | 1e-7 | 1e-7 | 1e-7 |
| LR warmup iters | 2000 | 500 | 500 |
| Sequence length | 2048 | 2048 | 2048 |
| Batch size (tokens) | 4M | 4M | 4M |
| $\beta$ for $\mathcal{L}_b$ | 0.01 | 0.01 | 0.01 |
| Tensor parallel | 1 | 1 | 1 |
| Pipeline parallel | 1 | 1 | 1 |

## B.2 CONTINUE-TUNING LLAMA3-8B

**Training Hyper-Parameters.** Tab. D shows the detailed training hyper-parameters of MoH-LLaMA3-8B. We find that if there is a discrepancy between the continue-training data and the original training data distribution of the model, the performance of the model may fluctuate wildly at the beginning of the training process. Since we do not have access to the raw training data of LLaMA3, we address these potential performance fluctuations by dividing the training process into two stages. In the first stage, we continue-tune the original LLaMA3-8B model using 300B tokens to

adapt it to our dataset. In addition, during the first stage, to enhance the Chinese ability of the model, we expand the vocabulary size. Specifically, we increase the original LLaMA3-8B vocabulary size from 128,256 to 160,896. In the second stage, we continue-tune this adapted model into our proposed MoH model with 100B tokens. During the first stage, the maximum learning rate is set to 6e-5, and the final learning rate is 6e-6. In the second stage, the maximum learning rate is set to 2e-5, and the final learning rate is 1e-6. For both stages, we employ the AdamW optimizer (Loshchilov & Hutter, 2017), with a batch size of 16 million tokens with a sequence length of 8192. During training, we use a weight decay of 0.1 and gradient clipping of 1.0.

Table D: **Training hyper-parameters of MoH-LLaMA3-8B.** We divide the training process into two stages. In the first stage, we continue-tune the LLaMA3-8B model using 300B tokens. In the second stage, we continue-tune this adapted model into our proposed MoH model with 100B tokens.

|  | The First Stage | The Second Stage |
|---|---|---|
| Training budget | 300B | 100B |
| Maximum learning rate | 6e-5 | 2e-5 |
| Final learning rate | 6e-6 | 1e-6 |
| LR warmup iters | 50 | 50 |
| Sequence length | 8192 | 8192 |
| Batch size (tokens) | 16M | 16M |
| $\beta$ for $\mathcal{L}_b$ | - | 0.01 |
| Tensor parallel | 2 | 1 |
| Pipeline parallel | 1 | 8 |

Table E: **Comparisons between MoH-LLaMA3-8B and LLaMA3-8B-stage1.** MoH-LLaMA3-8B outperforms LLaMA3-8B-stage1 by utilizing only 75% of the attention heads.

| Methods | #Activated Heads (%) | MMLU (5) | CMMLU (5) | NQ (32) | GSM8K(8) | TruthfulQA |
|---|---|---|---|---|---|---|
| LLaMA3-8B-stage1 | 100 | 66.2 | 66.0 | 28.1 | 58.6 | 41.9 |
| **MoH-LLaMA3-8B** | 75 | 65.8 | 64.4 | 28.3 | 56.9 | 44.0 |

| Methods | #Activated Heads (%) | HellaSwag (10) | LogiQA | BoolQ (32) | LAMBADA | SciQ |
|---|---|---|---|---|---|---|
| LLaMA3-8B-stage1 | 100 | 79.4 | 30.4 | 85.1 | 75.8 | 92.2 |
| **MoH-LLaMA3-8B** | 75 | 80.1 | 30.3 | 84.0 | 76.4 | 92.2 |

| Methods | #Activated Heads (%) | PIQA | WinoGrande | ARC-E | ARC-C (25) | Average |
|---|---|---|---|---|---|---|
| LLaMA3-8B-stage1 | 100 | 79.1 | 73.0 | 70.9 | 59.6 | 64.7 |
| **MoH-LLaMA3-8B** | 75 | 78.8 | 72.9 | 72.5 | 60.1 | **64.8** |

## C    ADDITIONAL EXPERIMENTS

**Comparison between MoH-LLaMA3-8B and LLaMA3-8B-stage1.**    We divide the training process into two stages. Tab. E shows the comparison between MoH-LLaMA3-8B and the model at the end of the first training stage (LLaMA3-8B-stage1). As shown in Tab. E, MoH-LLaMA3-8B quickly recovers the performance of LLaMA3-8B-stage1 within a training budget of 100B tokens. Notably, in English language tasks, MoH-LLaMA3-8B surpasses LLaMA3-8B-stage1 while using only 75% of the attention heads. However, for Chinese language and math tasks, the recovery performance of the MoH model is not as strong as for English. For example, MoH-LLaMA3-8B achieves an accuracy of 64.4% on CMMLU, compared to 66.0% for LLaMA3-8B-stage1. We attribute this to the fact that the model's Chinese and mathematical capabilities are primarily established during the first training stage. Since the first training stage uses only 300B tokens, significantly less than the 15T tokens in LLaMA3-8B's pre-training, the model's abilities in these areas are not fully stable. In the second training stage, after switching to the MoH model, the model experiences more significant

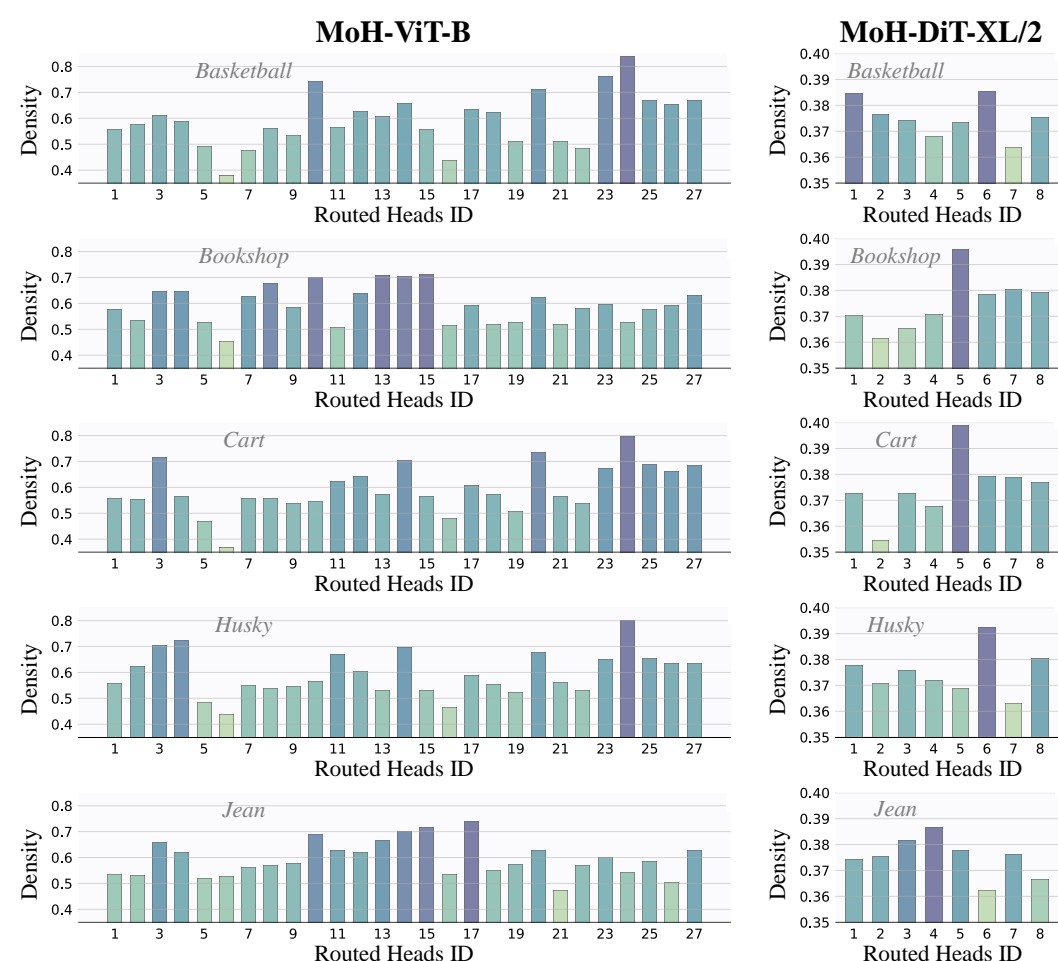

Figure A: **Additional visualization of the head load distribution in the final MoH layer.** For MoH-ViT-B and MoH-DiT-XL/2, we present the head load distributions for the categories "Basketball", "Bookshop", "Cart", "Husky", and "Jean". MoH-ViT-B activates 75% of the attention heads. MoH-DiT-XL/2 activates 90% of the attention heads.

forgetting in Chinese and math tasks. Overall, as shown in Tab. E, MoH-LLaMA3-8B achieves an average accuracy of 64.8% across 14 benchmarks, outperforming LLaMA3-8B-stage1 by utilizing only 75% of the attention heads.

**Effect of the Activated Head Ratio.** As shown in Tab. F, activating more attention heads generally leads to improved model performance. These results are intuitive, as activating more attention heads equates to utilizing more parameters and performing additional computations on the input.

Table F: **Ablation study on the impact of the activated head ratio.** All results are from MoH-ViT-S, by using a training budget of 100 epochs.

| Activated Heads | 50% | 55% | 60% | 65% | 70% | 75% | 80% |
|---|---|---|---|---|---|---|---|
| **Accuracy (%)** | 78.32 | 78.38 | 78.44 | 78.50 | 78.42 | 78.58 | **78.78** |

## D ADDITIONAL QUALITATIVE ANALYSIS

**Additional Visualization of the Head Load Distribution.** We provide additional visualization of the head load distribution in Fig. A. As illustrated in both Fig. 3 and Fig. A, there is notable variation in attention head assignments across different categories and task topics. This suggests that the MoH

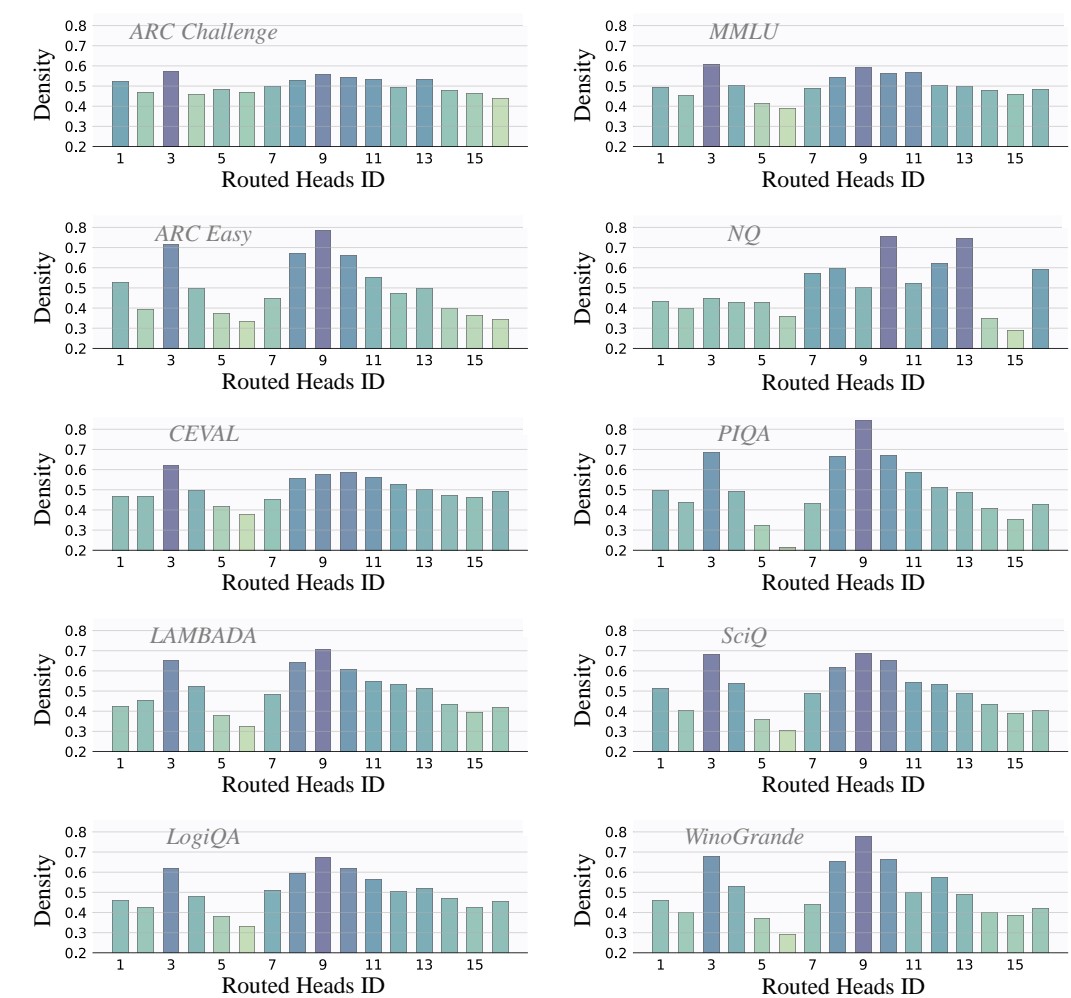

Figure B: **Additional visualization of the head load distribution in MoH-LLaMA3-8B.**

model adapts to a wide range of tasks by utilizing distinct head assignment patterns. This ability enables MoH to allocate attention heads more effectively to specific task types, leading to more efficient parameter utilization compared to standard multi-head attention.

**Additional Visualization of the Head Load Distribution in MoH-LLaMA3-8B.** We provide additional visualization of the head load distribution in Fig. B. As shown in Fig. B, MoH-LLaMA3-8B exhibits similar characteristics to MoH-LLMs trained from scratch, with significant variation in attention head assignments across different categories and task topics. This indicates that continue-tuning enables the model to adopt different head assignment patterns quickly. These results demonstrate that pre-trained multi-head attention models can be effectively continue-tuned into MoH models, significantly broadening the applicability of the proposed MoH approach.

**Additional Visualization of the Head Routing Score Distribution.** We provide additional visualization of the head routing score distribution in Fig. C, Fig. D, and Fig. E. As illustrated in Fig. C, Fig. D, and Fig. E, these head routing scores also vary across categories and task types. This dynamic weighting mechanism allows MoH to adjust the importance of each head in response to different task requirements, further enhancing its flexibility and performance. Besides, we find that the routing scores of shared heads change more across categories than those of routing headers. We consider this because routed heads adapt to different categories by adjusting their activation, while shared heads remain activated all the time. Therefore, shared heads primarily rely on changes in routing scores to adapt to different categories.

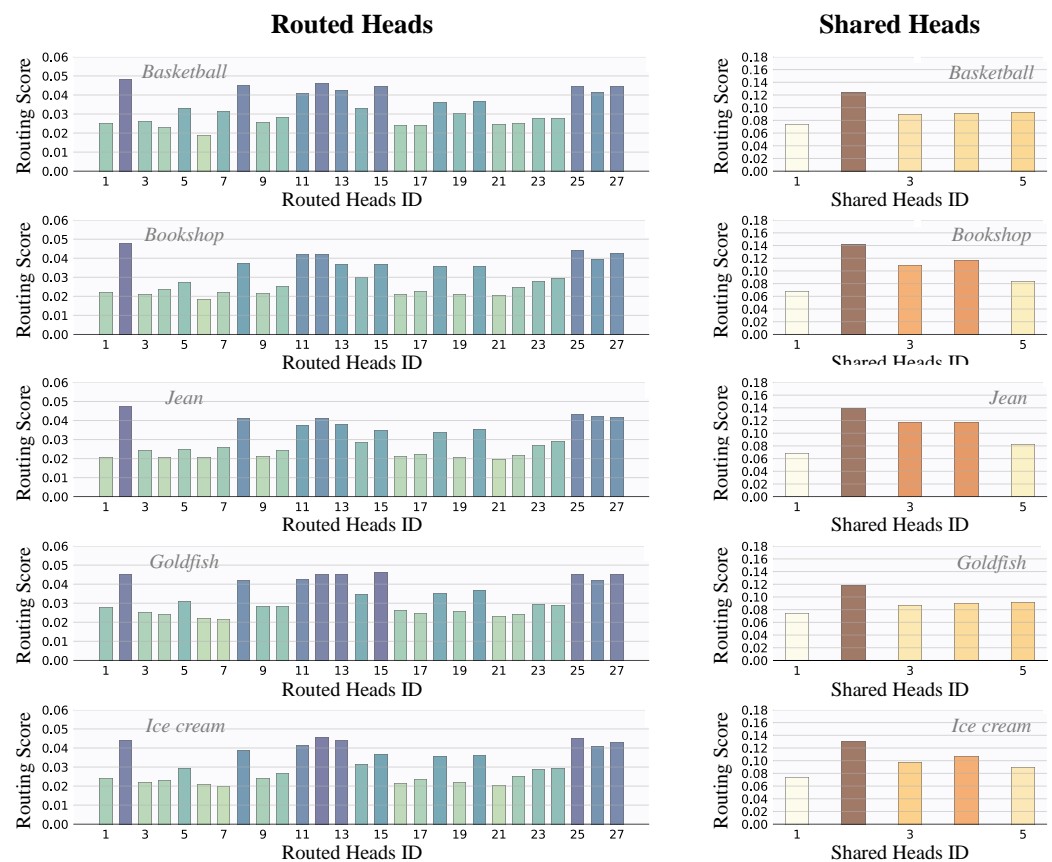

Figure C: **Additional visualization of the head routing score distribution in MoH-ViT-B.** MoH-ViT-B activates 75% of the attention heads.

**Images Generated from the Proposed MoH-DiT-XL/2 Model.** Fig. F shows samples generated by our class-conditional MoH-DiT-XL/2 model. These results demonstrate the ability of MoH-DiT-XL/2 to generate semantically correct content with accurate spatial relationships.

# E    DETAILS OF QUANTITATIVE EVALUATIONS FOR LLMs

We conduct comparative comparisons of MoH-LLM (MoH-LLaMA3-8B) against vanilla LLMs (LLaMA3-8B). The evaluation is performed on multiple key benchmarks using the Eleuther AI Language Model Evaluation Harness[§] (Gao et al., 2024), a unified framework for testing generative language models across a wide range of tasks. The benchmarks used for evaluation include:

**ARC** (Clark et al., 2018) is a multiple-choice question-answering resource featuring questions from science exams for grades 3 to 9. It is divided into two partitions: Easy and Challenge, with the latter containing more difficult questions that necessitate reasoning. Most questions offer four answer choices, while less than 1% feature either three or five choices. Additionally, ARC includes a supporting knowledge base with 14.3 million unstructured text passages. We report 0-shot accuracy on ARC Easy and 25-shot accuracy on ARC Challenge.

**LAMBADA** (Paperno et al., 2016) is an open-ended cloze task consisting of approximately 10,000 passages from BooksCorpus, where the objective is to predict a missing target word in the last sentence of each passage. The missing word is always the last word of the final sentence, with no options provided. We report 0-shot accuracy on LAMBADA.

---

[§]https://github.com/EleutherAI/lm-evaluation-harness

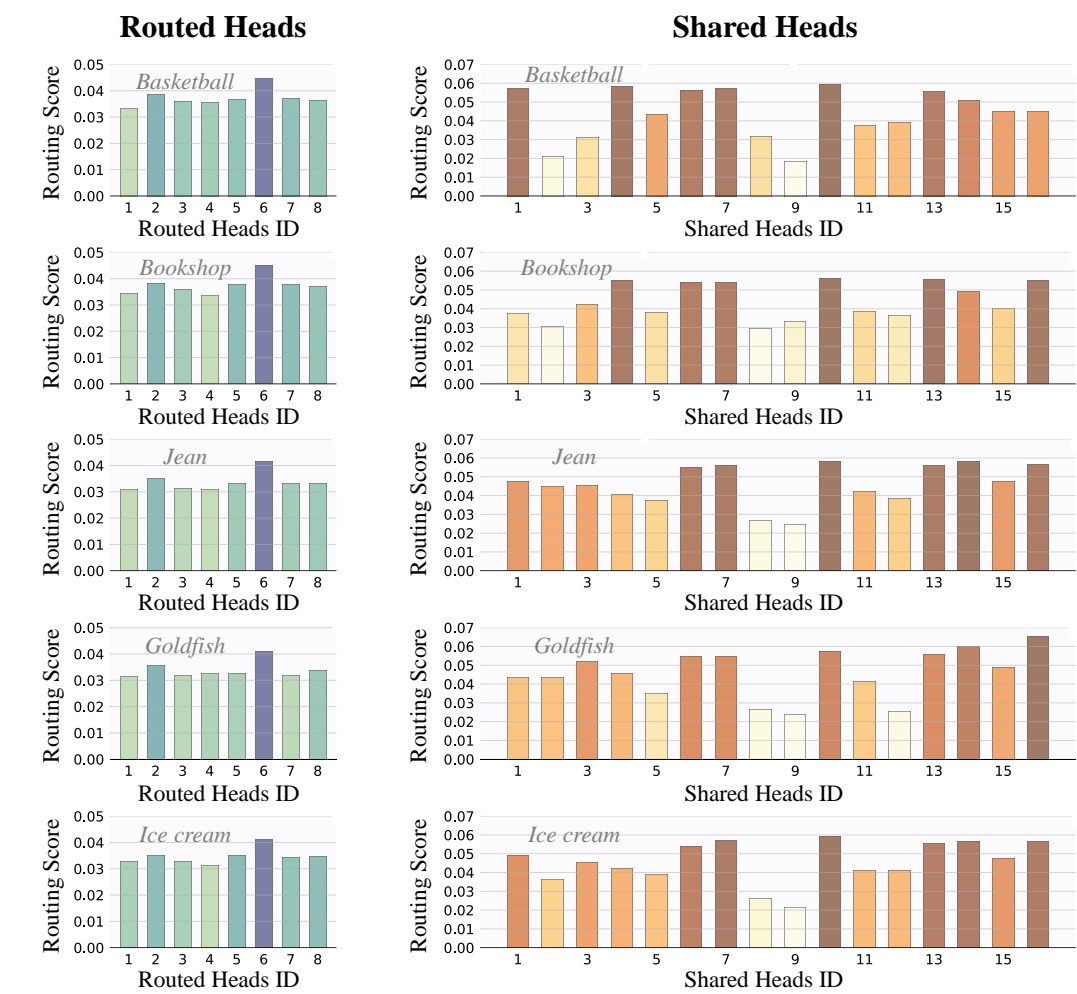

Figure D: **Additional visualization of the head routing score distribution in MoH-DiT-XL/2. MoH-DiT-XL/2 activates 90% of the attention heads.**

**LogiQA** (Liu et al., 2020) comprises 8,678 question-and-answer instances that encompass various types of deductive reasoning. The dataset serves as a benchmark for reexamining logical AI within the context of deep learning in NLP. We report 0-shot accuracy on LogiQA.

**PIQA** (Bisk et al., 2020) is a dataset designed for commonsense reasoning, aimed at evaluating the physical knowledge of current models. We report 0-shot accuracy on PIQA.

**SciQ** (Welbl et al., 2017) includes 13,679 crowdsourced science exam questions covering subjects such as Physics, Chemistry, and Biology. Each question is presented in a multiple-choice format with four answer options, and for most questions, an additional paragraph provides supporting evidence for the correct answer. We report 0-shot accuracy on SciQ.

**WinoGrande** (Sakaguchi et al., 2021) is a large-scale dataset comprising 44,000 problems, inspired by the original WSC design but enhanced to increase both its scale and difficulty. We report 0-shot accuracy on WinoGrande.

**HellaSwag** (Zellers et al., 2019) is a challenging dataset designed to evaluate commonsense natural language inference, which proves difficult for state-of-the-art models but poses no significant challenge for humans. We report the accuracy for the 10-shot HellaSwag.

**MMLU** (Hendrycks et al., 2021) is a benchmark designed to assess models' knowledge acquired during pretraining, making it more challenging and human-like in evaluation. It covers 57 subjects

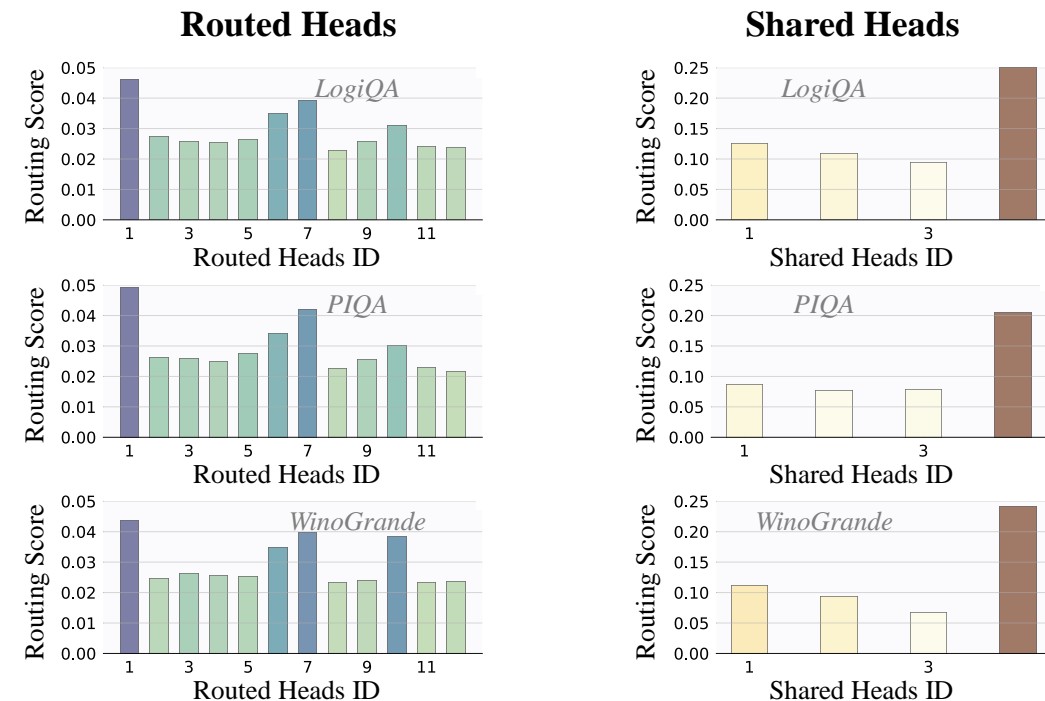

Figure E: **Additional visualization of the head routing score distribution in MoH-LLM-B.** MoH-LLM-B activate 75% of the attention heads.

across STEM, humanities, social sciences, and more, ranging from elementary to advanced professional levels. The benchmark tests both world knowledge and problem-solving skills, with subjects spanning traditional areas like math and history to specialized fields such as law and ethics, offering a comprehensive tool for identifying model blind spots. We report the accuracy for the 5-shot MMLU.

**Natural Questions (NQ)** (Kwiatkowski et al., 2019) is a question-answering dataset based on real, anonymized Google queries. Annotators label long and short answers (or null if no answer is found) from Wikipedia pages in the top 5 search results. The dataset includes 307,373 training examples, 7,830 development examples, and 7,842 test examples with 5-way annotations. We report the exact match score for 32-shot Natural Questions to measure the factual knowledge in the model.

**BoolQ** (Clark et al., 2019) is a question-answering dataset consisting of 15,942 yes/no questions. These questions are naturally occurring, and generated in unprompted and unconstrained contexts. Each example is provided as a triplet of (question, passage, and answer), with the page title optionally included as additional context. We report the accuracy for the 32-shot BoolQ.

**OpenbookQA** (Mihaylov et al., 2018) is a question-answering dataset designed to assess understanding of elementary-level science, similar to open-book exams. It contains 5,957 multiple-choice questions based on a "book" of 1,326 core science facts. The dataset requires not only knowledge of these facts but also the application of broad common knowledge. It includes mappings from each question to the core fact it targets and additional common knowledge facts. The dataset also provides scores of human accuracy and clarity, as well as crowd-sourced data for further analysis. We report 0-shot accuracy on OpenbookQA.

**TruthfulQA** (Lin et al., 2022) is a benchmark designed to evaluate the truthfulness of a language model's responses. It consists of 817 questions across 38 categories, such as health, law, finance, and politics. The questions are crafted to reflect common false beliefs or misconceptions that might lead humans to answer inaccurately. We report 0-shot accuracy on TruthfulQA.

**GSM8K** (Cobbe et al., 2021) is a dataset containing 8.5K high-quality, linguistically diverse grade school math word problems. It is divided into 7.5K training problems and 1K test problems. Each problem requires 2 to 8 steps to solve, typically involving a sequence of elementary calculations

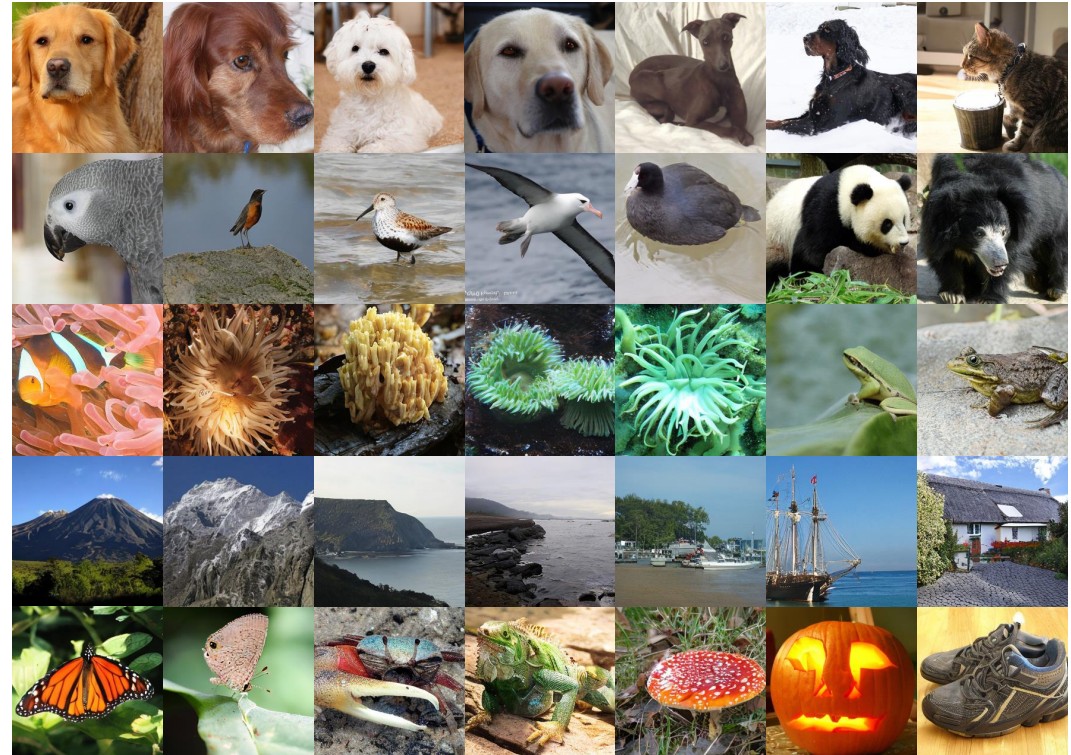

Figure F: **Images generated from the proposed MoH-DiT-XL/2 model.** We show samples generated from our class-conditional MoH-DiT-XL/2 model trained on ImageNet at 256×256 resolution. MoH-DiT-XL/2 activates 90% of the attention heads.

using basic arithmetic operations. A capable middle school student should be able to solve all the problems, making the dataset suitable for evaluating multi-step mathematical reasoning. We report the exact match score for 8-shot GSM8K.

**CEVAL** (Huang et al., 2023) is a comprehensive Chinese evaluation suite designed to assess the advanced knowledge and reasoning abilities of LLMs in a Chinese context. It includes multiple-choice questions across four difficulty levels (middle school, high school, college, and professional) and spans 52 diverse disciplines. We report the accuracy for the 5-shot CEVAL.

**CMMLU** (Li et al., 2023a) is a comprehensive Chinese benchmark designed to evaluate the knowledge and reasoning abilities of LLMs across various subjects, including natural sciences, social sciences, engineering, and humanities. We report the accuracy for the 5-shot CMMLU.

