# OpenReview forum: "MoH: Multi-Head Attention as Mixture-of-Head Attention"
_ICLR.cc/2025/Conference — Submitted to ICLR 2025_

### Official Review · Reviewer_3giJ · 2024-10-27

**Soundness:** 3
**Presentation:** 4
**Contribution:** 3
**Rating:** 6
**Confidence:** 4

**Summary:**

This paper observes that each attention head in multi-head attention operates in parallel. By formulating the multi-head attention in summation form, it builds a dynamic mixture-of-head attention without increasing the number of parameters. Meanwhile, it introduces shared heads and a two-stage routing mechanism to enhance the standard MoH method. Extensive experiments across popular model frameworks, including ViT, DiT, LLMs and continue-tuning LLMs demonstrate strong performance and applicability.

**Strengths:**

1. Without altering the number of parameters, this work treats standard multi-head attention as Mixture-of-Head attention, which enhances the flexibility of the attention mechanism and shows improved performance.
2. To consolidate common knowledge, it introduces shared heads along with a corresponding routing mechanism. The ablation study results have validated the effectiveness of these designs.
3. Popular tasks including ViT for image classification, DiT for image generation, and LLM for language generation, demonstrate superior performance. Furthermore, the proposed MoH attention can continue-tune pre-trained standard multi-head LLaMA3-8B, significantly enhancing its applicability.

**Weaknesses:**

1. This work claims enhanced inference efficiency multiple times. However, there is a lack of experimental evidence to support this claim. Upon reviewing the implementation provided in the Supplementary Material, it appears that you simply mask the useless heads after obtaining the whole multi-head results, which may not genuinely improve inference efficiency. Additionally, the process of dynamically routing each token to the appropriate heads could potentially increase inference costs.
2. The cited works that show multi-head attention contains redundant attention heads primarily focus on natural language processing, it would be better incorporate additional studies from the field of computer vision to provide a more comprehensive perspective.
3. In Equation 5 on line 190, the dimension of $W_r x_t $ is $h-h_s$, so for indices where $h_s+1<i \leq h$, $i$ in $(W_r x_t)_i$ should be
${i-h_s}$.
4. In Table 5 on line 383, does the baseline LLaMA3-8B refer to the model after continue-tuning with standard multi-head attention, consistent with the configuration of MoH models, or does it represent the starting point for continue-tuning? If the baseline is the starting point, it would be better to add a baseline that reflects the results after continue-tuning with multi-head attention, as continue-tuning is likely to improve performace.

**Questions:**

Please see weakness section

---

> ### Author Response · Authors · 2024-11-18
> **Responses to the Reviewer 3giJ [1/2]**
>
> We sincerely appreciate your thoughtful comments and your recognition of our method: "Without altering the number of parameters, this work treats standard multi-head attention as Mixture-of-Head attention, which enhances the flexibility of the attention mechanism and shows improved performance." Below, we provide detailed responses to your questions.
>
> **Q1: This work claims enhanced inference efficiency multiple times. However, there is a lack of experimental evidence to support this claim. Additionally, the process of dynamically routing each token to the appropriate heads could potentially increase inference costs.**
>
> **A1:** Thanks for your insightful comments. The code provided in the supplementary material does not genuinely improve inference efficiency.
>
> Implementing dynamic routing directly at the CUDA level would significantly enhance the efficiency of MoH. However, the support for MoE structures in the open-source community has remained limited, and developing CUDA operators independently is challenging for our small research team.
>
> Although we could not find an efficient open-source MoE routing framework, sparse matrix multiplication can serve as a temporary solution. Specifically, we transform the Q, K, and V features into sparse matrices using the mask generated by the router and replace the dense matrix multiplication in the attention mechanism with sparse matrix multiplication.
>
> As shown in the table below, although dynamic routing introduces additional computational overhead, MoH still outperforms standard multi-head attention mechanisms. Furthermore, as the input sequence gets longer, the advantage of MoH grows.
>
> |                      | # Head Num | # Head Dim | # Sequence Length | #Activated Heads (%) | Time (ms) |
> |:--------------------:|:----------:|:----------:|:-----------------:|:--------------------:|:---------:|
> | Multi-Head Attention |     32     |     64     |        256        |         100          |   0.360   |
> |         MoH          |     32     |     64     |        256        |          90          |   0.352   |
> |         MoH          |     32     |     64     |        256        |          75          |   0.321   |
> |       **MoH**        |     32     |     64     |        256        |          50          | **0.225** |
> | Multi-Head Attention |     32     |     64     |        512        |         100          |   1.376   |
> |         MoH          |     32     |     64     |        512        |          90          |   1.351   |
> |         MoH          |     32     |     64     |        512        |          75          |   1.180   |
> |       **MoH**        |     32     |     64     |        512        |          50          | **0.863** |
>
> It is worth noting that, to eliminate the impact of underlying operator optimizations, we replaced all matrix multiplications with sparse matrix multiplication when testing for speed.
>
> Using sparse matrix multiplication to implement MoH is only a temporary solution. The support for sparse matrix multiplication in CUDA operators is still very limited, as current GPUs are heavily optimized for dense matrix computations.
>
> We believe that as the open-source community continues to focus on MoE models, efficient MoE operators will be developed, enabling MoH to achieve even faster performance.
>
>
> **Q2: The cited works that show multi-head attention contains redundant attention heads primarily focus on natural language processing, it would be better incorporate additional studies from the field of computer vision to provide a more comprehensive perspective.**
>
> **A2:** Thanks for your insightful advice. Following your suggestion, we have expanded the discussion of related work in computer vision within the Introduction: "In computer vision, some works also identify attention head redundancy. Bhattacharyya et al. (2023) [1] reduces redundancy to boost performance, while Yun & Ro (2024) [2] develops single-head attention for efficiency."
>
>
> [1] Bhattacharyya, Mayukh, Soumitri Chattopadhyay, and Sayan Nag. "DeCAtt: Efficient Vision Transformers with Decorrelated Attention Heads." in CVPRW. 2023.
>
> [2] Yun, Seokju, and Youngmin Ro. "SHViT: Single-head vision transformer with memory efficient macro design." in CVPR. 2024.

---

> ### Author Response · Authors · 2024-11-18
> **Responses to the Reviewer 3giJ [2/2]**
>
> **Q3: In Equation 5 on line 190, the dimension of $W_{r}X_{t}$ is $h-h_s$, so for indices where $h_s + 1 < i \leq h$, $i$ in $\(W_{r}X_{t} \)_i $ should be $i-h_s$.**
>
> **A3:** Thank you for kindly pointing out our inappropriate indices $i$ in Equation 5. Following your suggestion, we have revised Equation 5 in the revised manuscript.
>
> Besides, when we carefully proofread the manuscript, we found that the indices in Equation 7 had the same issue. We have also modified Equation 7 in the revised manuscript.
>
>
> **Q4: In Table 5 on line 383, does the baseline LLaMA3-8B refer to the model after continue-tuning with standard multi-head attention, consistent with the configuration of MoH models, or does it represent the starting point for continue-tuning? If the baseline is the starting point, it would be better to add a baseline that reflects the results after continue-tuning with multi-head attention, as continue-tuning is likely to improve performace.**
>
> **A4:** The baseline LLaMA3-8B refer to the starting point for continue-tuning.
>
> Due to the page limit, we have included the performance of the model after continued training in Table E (Page 19) in the Appendix. Additionally, we have updated the caption of Table 5 to include the note, "Please refer to Tab. E in the Appendix for the performance of the model at the end of the first stage of training."
>
> As shown in the table below, MoH-LLaMA3-8B quickly recovers the performance of LLaMA3-8B-stage1 within a training budget of 100B tokens. Notably, in English language tasks, MoH-LLaMA3-8B surpasses LLaMA3-8B-stage1 while using only 75\% of the attention heads.
>
> However, for Chinese language and math tasks, the recovery performance of the MoH model is not as strong as for English.
> For example, MoH-LLaMA3-8B achieves an accuracy of 64.4\% on CMMLU, compared to 66.0\% for LLaMA3-8B-stage1.
> We attribute this to the fact that the model's Chinese and mathematical capabilities are primarily established during the first training stage.
> Since the first training stage uses only 300B tokens, significantly less than the 15T tokens in LLaMA3-8B's pre-training, the model's abilities in these areas are not fully stable.
> In the second training stage, after switching to the MoH model, the model experiences more significant forgetting in Chinese and math tasks.
>
> Overall, as shown in the table below, MoH-LLaMA3-8B achieves an average accuracy of 64.8\% across 14 benchmarks, outperforming LLaMA3-8B-stage1 by utilizing only 75\% of the attention heads.
>
> It is worth noting that continuing to train the MoH-LLaMA3-8B can further improve its performance.
> However, due to GPU resource constraints, we were limited to a training budget of 100B tokens.
>
>
> |             Methods             | # Activated Heads (%) | MMLU(5) | CMMLU(5) | NQ(32) | GSM8K(8) | TruthfulQA |
> |:-------------------------------:|:---------------------:|:-------:|:--------:|:------:|:--------:|:----------:|
> |         LLaMA3-8B-stage1        |          100          |  66.2   |   66.0   |  28.1  |   58.6   |    41.9    |
> |          MoH-LLaMA3-8B          |          75           |  65.8   |   64.4   |  28.3  |   56.9   |    44.0    |
>
> |             Methods             | # Activated Heads (%) | HellaSwag(10) | LogiQA | BoolQ(32) | LAMBADA | SciQ  |
> |:-------------------------------:|:---------------------:|:-------------:|:------:|:---------:|:-------:|:-----:|
> |         LLaMA3-8B-stage1        |          100          |     79.4      |  30.4  |   85.1    |  75.8   | 92.2  |
> |          MoH-LLaMA3-8B          |          75           |     80.1      |  30.3  |   84.0    |  76.4   | 92.2  |
>
> |             Methods             | # Activated Heads (%) | PIQA | WinoGrande | ARC-E | ARC-C(25) | Average  |
> |:-------------------------------:|:---------------------:|:----:|:----------:|:-----:|:---------:|:--------:|
> |         LLaMA3-8B-stage1        |          100          | 79.1 |    73.0    | 70.9  |   59.6    |   64.7   |
> |          MoH-LLaMA3-8B          |          75           | 78.8 |    72.9    | 72.5  |   60.1    | **64.8** |

---

> > ### Comment · Reviewer_3giJ · 2024-11-20
> > **Official Comment by Reviewer 3giJ**
> >
> > Thanks for addressing my concerns, I am in favor of acceptance.
> > > For example, MoH-LLaMA3-8B achieves an accuracy of 64.4% on CMMLU, compared to 66.0% for LLaMA3-8B-stage1. We attribute this to the fact that the model's Chinese and mathematical capabilities are primarily established during the first training stage.
> >
> > Is it possible to restore these capabilities by incorporating relevant data such as Chinese data during continued fine-tuning? I am somewhat apprehensive that failing to do so may impact the method's practical applicability. Additional ablation studies on this matter are welcomed.

---

> > > ### Author Response · Authors · 2024-11-21
> > > **Thank you for your invaluable feedback**
> > >
> > > We sincerely thank you for the time and effort you have devoted to providing comprehensive feedback on our responses.
> > >
> > > Following your suggestion, we added more Chinese data for continued fine-tuning, raising the probability of sampling Chinese data during training to 15%.
> > >
> > > As shown in the table below, the performance of the MoH-LLaMA3-8B model has not fully converged, and it continues to improve as the number of training tokens increases.
> > >
> > > |          Training Tokens            |  1B  |  2B  |  3B  |  4B  |  5B  |  6B  |  7B  |  8B  |  9B  | 10B  | 11B  | 12B  | 13B | 14B  | 15B  |   16B    |
> > > |:-----------------------------------:|:----:|:----:|:----:|:----:|:----:|:----:|:----:|:----:|:----:|:----:|:----:|:----:|:---:|:----:|:----:|:--------:|
> > > | MoH-LLaMA3-8B continued fine-tuning | 64.1 | 64.7 | 64.6 | 64.9 | 65.0 | 64.5 | 64.6 | 65.1 | 64.5 | 64.7 | 64.8 | 65.0 |64.7 | 64.9 | 64.8 | **65.3** |
> > >
> > >
> > > This aligns with our hypothesis that the model's Chinese capability is primarily developed during the first training stage. Since the first training stage uses only 300B tokens, significantly less than the 15T tokens in LLaMA3-8B's pre-training, the model's ability in this area is not fully stable. However, by increasing the training data, our proposed MoH model can also restore and enhance the capability of the original model.
> > >
> > >
> > > |                                           | # Activated Heads (%) | Training Tokens | CMMLU(5) |
> > > |:-----------------------------------------:|:---------------------:|:---------------:|:--------:|
> > > |                 LLaMA3-8B                 |          100          |       15T       |   50.7   |
> > > |             LLaMA3-8B-stage1              |          100          |      300B       |   66.0   |
> > > |               MoH-LLaMA3-8B               |          75           |      100B       |   64.4   |
> > > |       MoH-LLaMA3-8B continued fine-tuning |          75           |       16B       |   65.3   |
> > >
> > > **Thanks again for taking the time and effort when handling our paper.**

---

> > > ### Author Response · Authors · 2024-11-30
> > >
> > > Dear Reviewer
> > >
> > > Thank you so much for taking the time to review our work. We sincerely hope that our latest response has addressed your new question. If anything is still unclear or needs further explanation, we would be more than happy to provide additional details.
> > >
> > > As there hasn’t been much feedback from other reviewers yet, your insights are especially precious to us. Your feedback on our paper may have a direct impact on its acceptance.
> > >
> > > We sincerely appreciate your thoughtful feedback and efforts in helping us improve our work.

---

### Official Review · Reviewer_fpAk · 2024-11-02

**Soundness:** 3
**Presentation:** 2
**Contribution:** 2
**Rating:** 5
**Confidence:** 4

**Summary:**

The paper introduces Mixture-of-Head Attention (MoH) to the multi-head attention mechanism in Transformer models, incorporating a routing mechanism that activates the most relevant attention heads for each token. Extensive experiments across diverse model architectures demonstrate that MoH achieves comparable or better performance with fewer attention heads than traditional multi-head attention.

**Strengths:**

1. The idea of applying the mixture-of-experts paradigm to attention heads is novel.
2. MoH shows clear effectiveness for reducing computational overhead by activating fewer attention heads without sacrificing accuracy.
3. The paper presents a wide range of experiments across different model types, demonstrating the effectiveness of MoH. The ability to fine-tune pre-trained multi-head attention models like LLaMA adds practical value to the method.

**Weaknesses:**

1. The contribution is incremental. Replacing the summation of heads with a weighted sum and using expert selection are not entirely new ideas in machine learning, and their application here may not be sufficiently ground-breaking to warrant significant attention without a stronger theoretical basis.
2. The ablation studies are limited in scope and fail to deeply explore the design choices behind MoH. For example, there is little discussion on the impact of different numbers of activated heads beyond the experiments shown. The use of shared heads is also not well-motivated, and the reported improvements may be marginally due to tuning specific hyperparameters.
3. The paper could benefit from evaluations on more diverse and challenging tasks, such as object detection and instance segmentation, in line with prior research on ViT designs.
4. MoH introduces additional complexity with its routing mechanism. The added complexity is not fully justified by the performance gains, especially given that the gains appear marginal in some cases (e.g., DiT models).

**Questions:**

See Weaknesses.

---

> ### Author Response · Authors · 2024-11-18
> **Responses to the Reviewer fpAk [1/3]**
>
> We sincerely appreciate your thoughtful comments and for acknowledging that our method is "is novel," "shows clear effectiveness for reducing computational overhead by activating fewer attention heads without sacrificing accuracy," and validated through "a wide range of experiments across different model types." Below, we provide detailed responses to your questions.
>
>
> **Q1: Replacing the summation of heads with a weighted sum and using expert selection are not entirely new ideas in machine learning, and their application here may not be sufficiently ground-breaking to warrant significant attention without a stronger theoretical basis.**
>
> **A1:** Thanks for your insightful comments. We demonstrate that MoH is superior to vanilla multi-head attention from both **theoretical and experimental perspectives**.
>
> Specifically, MoH not only improves efficiency and model performance but also helps different attention heads to specialize better compared to multi-head attention.
>
> **From the theoretical perspective**, in standard multi-head attention, all heads use the same data, which can cause them to learn similar features. Many studies [1,2,3,4] have pointed out that there are redundant heads in multi-head attention.
>
> Given a minibatch of data $D$, the gradient of each attention head in multi-head attention can be written as $E_{x \in D} [ \frac{\partial \mathcal{L(x)}}{\partial h_{i} } ]$.
>
> In contrast, in MoH, routed heads are trained only on smaller subsets of data specifically assigned to them.
> In MoH's routing mechanism, the data is divided into $h-h_s$ subsets $D_1,D_2,...,D_{h-h_s}$, with each subset corresponding to a routed head. Besides, the routing score for each attention head acts as an adaptive adjustment to the learning rate, enabling the attention heads in MoH to specialize more effectively.
>
> Given a minibatch of data $D$ and the router $G(*)$, the gradient of each routed head in MoH can be written as $E_{x \in D_i } [G(x) \frac{\partial \mathcal{L(x)}}{\partial h_{i} }] $.
> The gradient of each shared head in MoH can be written as $E_{x \in D } [G(x) \frac{\partial \mathcal{L(x)}}{\partial h_{i} }] $.
>
> As shown in the table below, the routing mechanism and adaptive weights in MoH enable attention heads to specialize more effectively compared to standard multi-head attention.
>
> |                      | # Head Type |  # Data   | # Weight (learning rate) |                                                                               # Gradient                                                                                |
> |:--------------------:|:-----------:|:---------:|:-----------:|:-----------------------------------------------------------------------------------------------------------------------------------------------------------------------:|
> | Multi-Head Attention |      -      |     $D$     |      1      |                                              $E_{x \in D} [\frac{\partial \mathcal{L(x)}}{\partial h_{i} }] $                                               |
> |         MoH          | routed head | $D_i \subseteq D$ |    $G(x)$     |                                          $E_{x \in D_i } [G(x) \frac{\partial \mathcal{L(x)}}{\partial h_{i} }] $                                           |
> |         MoH          | shared head |     $D$     |    $G(x)$     |                                           $E_{x \in D } [G(x) \frac{\partial \mathcal{L(x)}}{\partial h_{i} }] $                                            |
>
>
> **From the experimental perspective**, we calculated the similarity of attention patterns and output features of different attention heads (include routed heads and shared heads).
>
> As shown in the table below, the similarity of attention patterns and output features among attention heads in MoH is lower than in standard multi-head attention, indicating reduced redundancy and greater differentiation among the attention heads in MoH.
>
> |                      | Similarity of Attention Patterns |            | Cosine Similarity of Output Features |            |
> |:--------------------:|:--------------------------------:|:----------:|:------------------------------------:|:----------:|
> |                      |               ViT                |    LLM     |                 ViT                  |    DiT     |
> | Multi-Head Attention |              0.5159              |   0.4795   |                0.0411                |   0.2550   |
> |       **MoH**        |            **0.3978**            | **0.4333** |              **0.0165**              | **0.2042** |
>
> *Given a pair of attention score matrices A and A', we calculate the similarity of attention patterns as $ 1 - \frac{1}{2} E_{query}[ ||A-A'||_{1} ] $. Since attention scores form a probability distribution for each query, the similarity is always between 0 to 1.*

---

> ### Author Response · Authors · 2024-11-18
> **Responses to the Reviewer fpAk [2/3]**
>
> [1] Voita, Elena, et al. "Analyzing multi-head self-attention: Specialized heads do the heavy lifting, the rest can be pruned." in ACL. 2019.
>
> [2] Yun, Seokju, and Youngmin Ro. "SHViT: Single-head vision transformer with memory efficient macro design." in CVPR. 2024.
>
> [3] Michel, Paul, Omer Levy, and Graham Neubig. "Are sixteen heads really better than one?." in NeurIPS. 2019.
>
> [4] Bhattacharyya, Mayukh, Soumitri Chattopadhyay, and Sayan Nag. "DeCAtt: Efficient Vision Transformers with Decorrelated Attention Heads." in CVPRW. 2023.
>
> **Q2: There is little discussion on the impact of different numbers of activated heads.**
>
> **A2:** Thanks for your helpful advice. As suggested, we conducted ablation experiments and updated the results in the revised manuscript (Tab.F on Page 20).
>
> The table below shows that activating more attention heads generally leads to improved model performance. These results are intuitive, as activating more attention heads equates to utilizing more parameters and performing additional computations on the input.
>
> | Activated Heads (%) |  50   |  55    |   60   |  65   |  70   |   75   |    80     |
> |:-------------------:|:-----:|:------:|:------:|:-----:|:-----:|:------:|:---------:|
> |    Accuracy (%)     | 78.32 | 78.38 | 78.44 | 78.50 | 78.42 | 78.58  | **78.78** |
>
>
> **Q3: The use of shared heads is not well-motivated.**
>
> **A3:** Thanks for your thoughtful comments. We introduce shared heads to efficiently learn common knowledge.
>
> In attention mechanism, some attention heads may capture common knowledge across different contexts, such as grammatical rules in language. Therefore, we designate a subset of heads as shared heads that remain always activated.
> By consolidating common knowledge within shared heads, we reduce redundancy among the other dynamically routed heads.
>
> We analyzed the feature rank of shared heads and routed heads. A lower feature rank indicates a higher correlation between features of different samples, suggesting that the features capture more general knowledge.
>
> As shown in the table below, the feature rank of shared heads is significantly lower than that of routed heads. This suggests that shared heads primarily capture common information across samples, while routed heads focus on information unique to individual samples.
>
> |     | Hidden Size | **Feature Rank of Shared Heads** | **Feature Rank of Routed Heads** |
> |:---:|:-----------:|:--------------------------------:|:--------------------------------:|
> | ViT |     768     |             **164**              |               270                |
> | LLM |    1536     |             **1123**             |               1441               |
>
> In summary, the combination of shared heads and routed heads allows MoH to effectively handle both general knowledge and class-specific knowledge.
>
>
> **Q4: The reported improvements may be marginally due to tuning specific hyperparameters.**
>
> **A4:** To ensure a fair comparison, we only replace the standard multi-head attention with the MoH, while keeping all other training parameters identical to ViT, DiT, and LLM. In other words, the training parameters were fully optimized for multi-head attention, and no additional parameter tuning was performed for the MoH model. Despite no additional parameter tuning, MoH outperforms multi-head attention by using only 50\%$\sim$90\% of the attention heads.
>
>
> **Q5: The paper could benefit from evaluations on more diverse and challenging tasks, such as object detection and instance segmentation, in line with prior research on ViT designs.**
>
> **A5:** Thanks for your insightful advice. Following your suggestion, we conducted experiments on object detection and instance segmentation tasks. Specifically, we used a Mask R-CNN detection head, trained with a 1× schedule, to evaluate the performance of ImageNet-1K pretrained MoH-ViT-S on the COCO dataset.
>
> As shown in the table below, MoH-ViT-S shows competitive performance in both object detection and instance segmentation.
>
> |     | # Activated Heads (%) | **Box mAP** | **Mask mAP** |
> |:---:|:---------------------:|:-----------:|:------------:|
> | Swin-S |          100          |    44.8     |     40.9     |
> | CSWin-S |          100          |    47.9     |     43.2     |
> | SG-Former-M |          100          |    48.2     |     43.6     |
> | VMambaV9-S |           -           |    48.7     |     43.7     |
> | MoH-ViT-S |          80           |  **51.0**   |   **45.5**   |

---

> ### Author Response · Authors · 2024-11-18
> **Responses to the Reviewer fpAk [3/3]**
>
> **Q6: MoH introduces additional complexity with its routing mechanism.**
>
> **A6:** Thanks for your thoughtful comments. In our experiment settings, the additional complexity of routing mechanism is small, especially for long input sequences.
>
> In MoH, we transform the Q, K, and V features into sparse matrices using the mask generated by the router and replace the dense matrix multiplication in the attention mechanism with sparse matrix multiplication.
>
> As shown in the table below, although dynamic routing introduces additional computational overhead, MoH still outperforms standard multi-head attention mechanisms. Moreover, as the input sequence length increases, the extra overhead impact from routing mechanism decreases.
>
> |                      | # Head Num | # Head Dim | # Sequence Length | #Activated Heads (%) | Time Including Routing (ms) | Routing Time (ms) |
> |:--------------------:|:----------:|:----------:|:-----------------:|:--------------------:|:---------------------------:|:-----------------:|
> | Multi-Head Attention |     32     |     64     |        256        |         100          |            0.360            |         -         |
> |         MoH          |     32     |     64     |        256        |          90          |            0.352            |       0.113       |
> |         MoH          |     32     |     64     |        256        |          75          |            0.321            |       0.086       |
> |       **MoH**        |     32     |     64     |        256        |          50          |          **0.225**          |       0.066       |
> | Multi-Head Attention |     32     |     64     |        512        |         100          |            1.376            |         -         |
> |         MoH          |     32     |     64     |        512        |          90          |            1.351            |       0.126       |
> |         MoH          |     32     |     64     |        512        |          75          |            1.180            |       0.093       |
> |       **MoH**        |     32     |     64     |        512        |          50          |          **0.863**          |       0.083       |
>
> To eliminate the impact of underlying operator optimizations, we replaced all matrix multiplications with sparse matrix multiplication when testing for speed.
>
> Using sparse matrix multiplication to implement MoH is only a temporary solution. Implementing dynamic routing directly at the CUDA level would make MoH significantly faster. However, for our small research team, independently developing CUDA operators is challenging.
>
> We believe that as the open-source community continues to focus on MoE models, efficient MoE operators will be developed, enabling MoH to achieve even faster performance.
>
> **Q7: The added complexity is not fully justified by the performance gains, especially given that the gains appear marginal in some cases (e.g., DiT models).**
>
> **A7:** Thanks for your insightful comments. In fact, as shown in the table below, our MoH-DiT-XL/2 achieves significant improvements over DiT.
>
> |                             | # Activated Heads (%) |   FID    |   sFID   |     IS     |   Precision   |  Recall  |
> |:---------------------------:|:---------------------:|:--------:|:--------:|:----------:|:-------------:|:--------:|
> |          DiT-XL/2           |          100          |   9.62   |   6.85   |   121.50   |     0.67      | **0.67** |
> |     DiT-XL/2 (cfg=1.25)     |          100          |   3.22   |   5.28   |   201.77   |     0.76      |   0.62   |
> |      **MoH-DiT-XL/2**       |          90           |   8.56   |   6.61   |   129.54   |     0.68      | **0.67** |
> | **MoH-DiT-XL/2 (cfg=1.25)** |          90           | **2.94** | **5.17** | **207.25** |   **0.77**    |   0.63   |

---

> ### Author Response · Authors · 2024-11-22
>
> Dear Reviewer
>
> Could we kindly inquire if the responses have satisfactorily tackled your concerns, or if there is a need for further clarification? Your commitment to reviewing our work is immensely appreciated, and we express our sincere gratitude for your insightful comments and the considerable time you have dedicated to reviewing our paper.

---

> ### Author Response · Authors · 2024-12-01
> **A gentle reminder**
>
> Dear Reviewer
>
> Thank you once again for your thorough review. We hope that our responses and the revised manuscript have effectively addressed your concerns. We would greatly appreciate your feedback. Please feel free to let us know if you have any further questions.

---

### Official Review · Reviewer_Z5Ku · 2024-11-04

**Soundness:** 3
**Presentation:** 3
**Contribution:** 3
**Rating:** 6
**Confidence:** 3

**Summary:**

The authors introduce an approach that mitigates redundancy among attention heads through the use of the MoE, which adaptively selects attention heads according to input tokens. This method enhances inference efficiency by employing only those heads that are crucial for feature extraction during the inference process. The MoH demonstrates enhanced performance, even when utilizing a limited number of heads, as evidenced by comprehensive validation experiments.


I appreciate the authors' response. The response provided to my comment appears to be satisfactory, so I keep my score.

**Strengths:**

- The author conducts comprehensive verification experiments to assess the performance of MoH, demonstrating results that are equal to or surpass previous methods.
- MoH can significantly reduce the head's resources, which can tackle the most important problem of heavy MHSA operations.

**Weaknesses:**

- The author performed ablation studies utilizing different ratios to determine the optimal configuration of shared heads or activated heads; however, this approach is heuristic. The ablation study concerning the ratio of shared heads presented in Table 7 indicates that identifying the optimal head ratio shows significant challenges.
- Given that the primary focus of this paper is to enhance inference efficiency through the reduction of multi-head ratios, it is essential to conduct an experiment that compares this approach with prior methods aimed at decreasing multi-head ratios.
- The author asserts that the shared head acquires common knowledge in Line [180-183], yet this paper is limited to providing evidence that the shared head genuinely learns common knowledge.

**Questions:**

The author claims that the inference efficiency is improved by MoH. However, there is limited ground for this claim in the experiment. Is there any more evidence to support this claim? If further experiments are difficult, even a theoretical interpretation should be presented.

---

> ### Author Response · Authors · 2024-11-18
> **Responses to the Reviewer Z5Ku [1/2]**
>
> We sincerely appreciate your thoughtful comments, especially noting that we "conduct comprehensive verification experiments" and our method "can tackle the most important problem of heavy MHSA operations." Below, we provide detailed responses to your questions.
>
> **Q1: Table 7 indicates that identifying the optimal head ratio shows significant challenges.**
>
> **A1:** Thanks for your insightful comments. Identifying the optimal head ratio remains a challenge that requires further exploration.
> However, determining the optimal hyper-parameters of a model is a common challenge in deep learning.
>
>
> For example, even in multi-head self-attention mechanisms, finding the optimal number of heads and their dimensions is not a fully solved problem. As an illustration, increasing the number of heads in a ViT model from 24 to 32 (while reducing the corresponding head dimensions from 32 to 24) improves its accuracy by 0.1 on the ImageNet-1K classification.
>
>
> We hope future research will provide more insights into determining optimal hyper-parameters for our method.
>
> **Q2: A comparison of this approach with prior methods aimed at decreasing multi-head ratios.**
>
> **A2:** Thanks for your valuable advice. As suggested, we trained a new MoH model with a similar number of parameters as SHViT [1] and compared its performance to SHViT. Unlike our method, SHViT uses convolution and single-head attention to address head redundancy in ViT.
>
>
> As shown in the table below, our MoH model significantly outperforms SHViT in overall performance.
>
> |             |        Params (M)        | **Top-1 Acc (%)** | **Top-5 Acc (%)** |
> |:-----------:|:------------------------:|:-----------------:|:-----------------:|
> |  SHViT-S1   |         11.4          |       75.2        |       92.4        |
> |  SHViT-S2   |         14.2          |       77.4        |       93.4        |
> |  SHViT-S3   |         16.5          |       79.4        |       94.5        |
> | **MoH-ViT** |         12.9          |     **82.1**      |     **96.1**      |
>
>
> [1] Yun, Seokju, and Youngmin Ro. "SHViT: Single-head vision transformer with memory efficient macro design." in CVPR. 2024.
>
>
> **Q3: Evidence that the shared head genuinely learns common knowledge.**
>
> **A3:** Thanks for your helpful comments. We provide both theoretical and experimental evidence that shared heads learn common knowledge.
>
> * **From the theoretical perspective**, routed heads are trained only on smaller subsets of data specifically assigned to them, while shared heads are trained on the entire dataset. In MoH's routing mechanism, the dataset is divided into $h-h_s$ subsets ${D_1,D_2,...,D_{h-h_s}}$, with each subset corresponding to a routed head. In contrast, shared heads are trained on the full dataset, $D=D_1 \cup D_2 \cup ... \cup D_{h-h_s}$. Thus, shared heads capture the common knowledge across the entire dataset, while routed heads focus on the specific knowledge of their respective subsets.
>
> * **From the experimental perspective**, we analyzed the feature rank of shared heads and routed heads. A lower feature rank indicates a higher correlation between features of different samples, suggesting that the features capture more general knowledge. As shown in the table below, the feature rank of shared heads is significantly lower than that of routed heads. This suggests that shared heads primarily capture common information across samples, while routed heads focus on information unique to individual samples.
>
> |     | Hidden Size | **Feature Rank of Shared Heads** | **Feature Rank of Routed Heads** |
> |:---:|:-----------:|:--------------------------------:|:--------------------------------:|
> | ViT |     768     |             **164**              |               270                |
> | LLM |    1536     |             **1123**             |               1441               |
>
> In summary, whether from theoretical intuition or experimental results, shared heads are more tented to capture general knowledge compared to routed heads.

---

> ### Author Response · Authors · 2024-11-18
> **Responses to the Reviewer Z5Ku [2/2]**
>
> **Q4: Is there any more evidence to support the inference efficiency is improved by MoH? If further experiments are difficult, even a theoretical interpretation should be presented.**
>
> **A4:** Thanks for your insightful advice. Since MoE models gained significant attention following the rise of LLMs, the support for MoE structures in the open-source community has remained limited.
> While we could not find an efficient open-source MoE routing framework, sparse matrix multiplication can serve as a temporary solution.
>
> Specifically, we transform the Q, K, and V features into sparse matrices using the mask generated by the router and replace the dense matrix multiplication in the attention mechanism with sparse matrix multiplication.
>
> As shown in the table below, although dynamic routing introduces additional computational overhead, MoH still outperforms standard multi-head attention mechanisms.
> Furthermore, as the input sequence gets longer, the advantage of MoH grows.
>
> |                      | # Head Num | # Head Dim | # Sequence Length | #Activated Heads (%) | Time (ms) |
> |:--------------------:|:----------:|:----------:|:-----------------:|:--------------------:|:---------:|
> | Multi-Head Attention |     32     |     64     |        256        |         100          |   0.360   |
> |         MoH          |     32     |     64     |        256        |          90          |   0.352   |
> |         MoH          |     32     |     64     |        256        |          75          |   0.321   |
> |       **MoH**        |     32     |     64     |        256        |          50          | **0.225** |
> | Multi-Head Attention |     32     |     64     |        512        |         100          |   1.376   |
> |         MoH          |     32     |     64     |        512        |          90          |   1.351   |
> |         MoH          |     32     |     64     |        512        |          75          |   1.180   |
> |       **MoH**        |     32     |     64     |        512        |          50          | **0.863** |
>
> It is worth noting that, to eliminate the impact of underlying operator optimizations, we replaced all matrix multiplications with sparse matrix multiplication when testing for speed.
>
> Using sparse matrix multiplication to implement MoH is only a temporary solution. Implementing dynamic routing directly at the CUDA level would make MoH significantly faster. However, for our small research team, independently developing CUDA operators is challenging.
>
> We believe that as the open-source community continues to focus on MoE models, efficient MoE operators will be developed, enabling MoH to achieve even faster performance.

---

> ### Author Response · Authors · 2024-11-22
>
> Dear Reviewer
>
> Would it be possible for us to kindly ascertain if the provided responses have satisfactorily tackled any concerns you may have had and if further explanations or clarifications are needed? Your generous investment of time and effort in the evaluation of our work is truly commendable. We extend our heartfelt gratitude for your insightful commentary and the considerable time you have devoted to reviewing our paper.

---

> ### Author Response · Authors · 2024-12-02
> **A gentle reminder**
>
> Dear Reviewer
>
> We truly appreciate your time and effort in helping us improve our work. We hope our responses have addressed your concerns. If anything is still unclear or needs more explanation, we are happy to provide further details. Since the discussion period ends soon, your comments are especially important to us.
>
> Thanks again for taking the time and effort when handling our paper.

---

### Official Review · Reviewer_Tm7Z · 2024-11-04

**Soundness:** 3
**Presentation:** 3
**Contribution:** 3
**Rating:** 6
**Confidence:** 3

**Summary:**

In the field of deep learning, multi-head attention mechanism has always been a core component of Transformer models, achieving great success in natural language processing and computer vision tasks. However, research has shown that not all attention heads are equally important, and many attention heads can be pruned without affecting model accuracy. Based on this insight, this paper proposes a new architecture called Mixture of Head Attention (MoH) aimed at improving the efficiency of attention mechanisms while maintaining or surpassing previous accuracy levels.

**Strengths:**

MoH can achieve competitive performance while using fewer attention heads.By introducing shared heads and a two-stage routing mechanism, MoH enhances the standard Mixture-of-Experts (MoE) method, enabling the model to capture shared knowledge more effectively across different contexts.

MoH can be fine-tuned from pre-trained multi-head attention models, such as LLaMA3-8B, significantly enhancing the applicability of the model.

The method has been validated across various popular model frameworks, including Vision Transformers (ViT), Diffusion Models (DiT), and Large Language Models (LLMs), demonstrating superior performance in both image classification and language tasks.

**Weaknesses:**

It is suggested to provide more evidence about the diversity within the selected heads. Visualizations and statistics of the distribution may provide more insights.

What is the effect of MoH on multi-task joint learning. More discussions or experiments are welcomed.

What is the `density' in Figure.3. Is it a weight used to select whether to activate？

The discussion section indicate that MoA only suitable for encoder-decoder architecture.  It requires more evidence and explanation.

**Questions:**

Please see weaknesses.

---

> ### Author Response · Authors · 2024-11-18
> **Responses to the Reviewer Tm7Z [1/2]**
>
> We sincerely appreciate your thoughtful comments and for recognizing that our method "achieves competitive performance with fewer attention heads," "can be fine-tuned from pre-trained multi-head attention models," and "has been validated across various popular model frameworks." Below, we address your questions in detail.
>
> **Q1: It is suggested to provide more evidence about the diversity within the selected heads. Visualizations and statistics of the distribution may provide more insights.**
>
> **A1:** Thanks for your insightful advice. As suggested, we have included more visualizations in the revision. Specifically, in the appendix of the revised manuscript, we have provided:
> * Additional visualizations of the head load distribution for MoH-ViT (Fig. A left on Page 20), MoH-DiT (Fig. A right on Page 20), and MoH-LLaMA3 (Fig. B on Page 21).
> * Additional visualizations of the head routing score distribution for MoH-ViT (Fig. C on Page 22), MoH-DiT (Fig. D on Page 23), and MoH-LLM (Fig. E on Page 24).
>
> **On the one hand**, as shown in the additional visualizations of the head load distribution, there is notable variation in attention head assignments across different categories and task topics. This suggests that the MoH model adapts to a wide range of tasks by utilizing distinct head assignment patterns. This ability enables MoH to allocate attention heads more effectively to specific task types, leading to more efficient parameter utilization than standard multi-head attention.
>
> **On the other hand**, MoH replaces the standard summation in multi-head attention with a weighted summation. As illustrated in the additional visualizations of the head routing score distribution, these head routing scores also vary across categories and task types. This dynamic weighting mechanism allows MoH to adjust the importance of each head in response to different task requirements, further enhancing its flexibility and performance.
>
> Besides, we also calculated the Coefficient of Variation (CV) of the head routing scores across categories and task topics.
> As shown in the table below, the routing scores of shared heads change more across categories than those of routing headers.
> We consider this because routed heads adapt to different categories by adjusting their activation, while shared heads remain activated all the time. Therefore, shared heads primarily rely on changes in routing scores to adapt to different categories.
>
>
> |                 |          **ViT**          |          **DiT**           |          **LLM**         |
> |:------------:|:------------------------:|:------------------------:|:------------------------:|
> |              | Coefficient of Variation | Coefficient of Variation | Coefficient of Variation |
> | Shared Heads |         11.913%          |          9.389%          |         12.777%          |
> | Routed Heads |         10.926%          |          4.741%          |          3.478%          |

---

> ### Author Response · Authors · 2024-11-18
> **Responses to the Reviewer Tm7Z [2/2]**
>
> **Q2: What is the effect of MoH on multi-task joint learning. More discussions or experiments are welcomed.**
>
> **A2:** Thanks for your valuable comments. We conducted two experiments:
>
> * Training large language models is naturally a form of multi-task learning because the training data comes from various sources, including CommonCrawl, C4, Github, Wikipedia, Books, ArXiv, StackExchange, and so on. As shown in the table below, MoH achieves higher average accuracy across multiple tasks compared to the multi-head attention model.
>
> |              | # Activated Heads (%) | **SciQ** | **PIQA** | **WinoGrande** | **OpenbookQA** | **LogiQA** | **TruthfulQA** | **Avg.** |
> |:------------:|:---------------------:|:--------:|:--------:|:--------------:|:--------------:|:----------:|:--------------:|:--------:|
> | LLM-B |          100          |   73.1   | **70.3** |      53.3      |    **32.4**    |    29.0    |      29.5      |   47.9   |
> | MoH-LLM-B |          75           | **76.0** |   69.2   |      52.7      |      30.4      |  **29.8**  |      32.6      | **48.5** |
> | MoH-LLM-B |          50           |   75.6   |   66.9   |    **53.5**    |      29.4      |    26.7    |    **32.7**    |   47.5   |
>
> * We combined several task-specific datasets (ShareGPT, WizardLM_evol_instruct, and SlimOrca for instruction following, MetaMath for math, and Evol-CodeAlpaca for code). Then, we fine-tuned LLaMA3-8B using this multi-task dataset. As shown in the table below, MoH also achieves higher average accuracy across multiple tasks compared to the multi-head attention model.
>
> |          | # Activated Heads (%) | **Language Tasks** |          |                |     **Math Task**      |    **Code Tasks**    |                       | **Avg.** |
> |:--------:|:---------------------:|:------------------:|:--------:|:--------------:|:----------------------:|:--------------------:|:---------------------:|:--------:|
> |          |                       |      **SciQ**      | **PIQA** | **WinoGrande** |   **GSM8K (8-shot)**   | **HumanEval pass@1** | **HumanEval pass@10** |          |
> | Baseline |          100          |        90.3        | **79.2** |    **73.7**    |        **64.7**        |         23.9         |         38.4          |   61.7   |
> | **MoH**  |         87.5          |      **92.3**      | **79.2** |      72.6      |          64.6          |       **25.4**       |       **40.2**        | **62.4** |
>
> **Q3: What is the "density" in Figure 3. Is it a weight used to select whether to activate?**
>
> **A3:** In Figure 3, "density" denotes the ratio of head activations to the total number of tokens. For example, if a head is activated 128 times out of 256 tokens, the density would be 0.5.
>
> To make Figure 3 easier to understand, we have added an explanation of "density" in the caption of Figure 3: "'density' denotes the ratio of the number of head activations to the total number of tokens."
>
>
> **Q4: The discussion section indicate that MoA only suitable for encoder-decoder architecture. It requires more evidence and explanation.**
>
> **A4:** Thank you for kindly pointing out our inappropriate description. A more precise statement would be that the experiments of MoA are conducted only on the encoder-decoder architecture for language tasks.
>
> In the revised manuscript, we have revised the original sentence "MoA is only validated on the encoder-decoder architecture for language tasks" to "MoA is only validated for language tasks."

---

> > ### Comment · Reviewer_Tm7Z · 2024-11-26
> > **Thanks for the response**
> >
> > Thanks for the detailed information and results provided by the authors. The response has addressed most of my concerns.

---

> > > ### Author Response · Authors · 2024-11-26
> > > **Sincere Appreciation**
> > >
> > > Thank you for your helpful feedback. Your expertise and careful review have greatly improved our work. We truly appreciate the time and effort you took to give such a detailed review of our paper.

---

> ### Author Response · Authors · 2024-11-22
>
> Dear Reviewer
>
> May we kindly inquire if the provided responses have adequately addressed any questions you might have had? Does there remain a requirement for further explanations or clarifications? We wish to express our sincere gratitude for your meticulous evaluation and for generously investing a significant amount of your time in reviewing our paper. Your feedback would be greatly valued.

---

### Author Response · Authors · 2024-11-18
**Global Response**

We sincerely thank all PCs, SACs, ACs, and Reviewers for their time and efforts when handling our paper.

All reviewers appreciate the contributions of our method:
* All reviewers pointed out that **MoH achieves competitive or superior performance while utilizing fewer attention heads**.
* All reviewers commented that we conduct **a wide range of experiments across different model types**, including Vision Transformers (**ViT**), Diffusion Models (**DiT**), and Large Language Models (**LLMs**), demonstrating **superior performance in both image classification and language tasks**.
* Reviewers Tm7Z, fpAk, and 3giJ all highlighted that **the proposed MoH can continue-tune pre-trained standard multi-head LLaMA3-8B, significantly enhancing its applicability**.

As suggested by the reviewers, we have revised the manuscript as follows:
* Additional visualizations of the head load distribution for MoH-ViT (Fig. A left on Page 20), MoH-DiT (Fig. A right on Page 20), and MoH-LLaMA3 (Fig. B on Page 21).
* Additional visualizations of the head routing score distribution for MoH-ViT (Fig. C on Page 22), MoH-DiT (Fig. D on Page 23), and MoH-LLM (Fig. E on Page 24).
* The ablation study on the impact of the activated head ratio (Tab.F on Page 20).
* Comparison between MoH-LLaMA3-8B and LLaMA3-8B-stage1 (Tab.E on Page 19).
* Discussion of related works about redundant attention heads in computer vision in the Introduction.
* Modifying the inappropriate indices in Equation 5 and Equation 7.
* Modifying the inappropriate description of MoA.
* Modifying the caption of Table 5 and Figure 3.

We highlight all the modifications using red color.

---

### Author Response · Authors · 2024-11-25
**The Discussion Period Ends Soon**

Dear Reviewers,

We sincerely appreciate the invaluable feedback provided by reviewers. Since the discussion period ends soon (November 26, AoE), we would appreciate hearing whether our rebuttal has addressed your concerns. We would be happy to provide more input if needed.

Thanks for your attention!

Best regards,

Authors of Paper#2876

---

### Meta-Review · Area_Chair_kFSM · 2024-12-23

**Metareview:**

This paper investigates neural architecture design and proposes a new architecture called a Mixture of Head Attention aimed at improving the efficiency of attention mechanisms while maintaining accuracy. The manuscript was reviewed by four experts in the field. The recommendations are ("5: marginally below the acceptance threshold", 3 x "6: marginally above the acceptance threshold"). The reviewers raised several concerns regarding the paper, e.g., incremental technical contribution, unconvincing experimental evaluations, etc. Considering the reviewers' concerns, we regret that the paper cannot be recommended for acceptance at this time. The authors are encouraged to consider the reviewers' comments when revising the paper for submission elsewhere.

**Additional Comments On Reviewer Discussion:**

Reviewers mainly hold concerns regarding the incremental contribution (the proposed operations are not entirely new in ML and the application in CV is not sufficiently ground-breaking and significant, with no inspiration and no new insights) (Reviewer fpAk), unconvincing experimental evaluations (e.g., more technical designs are not explored, need to be evaluated on more challenging tasks with convincing performance and model complexity, missing efficiency comparisons) (Reviewer fpAk, 3giJ). The authors' rebuttal could not fully address the above-listed concerns.

---

### Decision · Program_Chairs · 2025-01-22

Reject